# The Promyelocytic Leukemia Protein facilitates human herpesvirus 6B chromosomal integration, immediate-early 1 protein multiSUMOylation and its localization at telomeres

**Vanessa Collin[1], Annie Gravel[1], Benedikt B. Kaufer[2], Louis Flamand[1,3]***

1 Division of Infectious Disease and Immunity, CHU de Québec Research Center, Quebec City, Quebec, Canada, 2 Institut für Virologie, Freie Universität Berlin, Berlin, Germany, 3 Department of microbiology, infectious disease and immunology, Faculty of Medicine, Université Laval, Quebec City, Québec, Canada

* Louis.Flamand@crchudequebec.ulaval.ca

**Data Availability Statement:** All relevant data are within the manuscript and its Supporting Information files.

## Abstract

Human herpesvirus 6B (HHV-6B) is a *betaherpesvirus* capable of integrating its genome into the telomeres of host chromosomes. Until now, the cellular and/or viral proteins facilitating HHV-6B integration have remained elusive. Here we show that a cellular protein, the promyelocytic leukemia protein (PML) that forms nuclear bodies (PML-NBs), associates with the HHV-6B immediate early 1 (IE1) protein at telomeres. We report enhanced levels of SUMOylated IE1 in the presence of PML and have identified a putative SUMO Interacting Motif (SIM) within IE1, essential for its nuclear distribution, overall SUMOylation and association with PML to nuclear bodies. Furthermore, using PML knockout cell lines we made the original observation that PML is required for efficient HHV-6B integration into host chromosomes. Taken together, we could demonstrate that PML-NBs are important for IE1 multiSUMOylation and that PML plays an important role in HHV-6B integration into chromosomes, a strategy developed by this virus to maintain its genome in its host over long periods of time.

## Author summary

Human herpesvirus 6B (HHV-6B) is a ubiquitous virus that can be life threatening in immunocompromised patients. HHV-6B is among a few other herpesviruses that integrate their genome in host chromosomes as a mean to establish dormancy. Integration of HHV-6B occurs in host telomeres, a region that protects our genome from deterioration and controls the cellular lifespan. To date, the mechanisms leading to HHV-6B integration remain elusive. Our laboratory has identified that the IE1 protein of HHV-6B associates with PML, a cellular protein that is responsible for the regulation of important cellular mechanisms including DNA recombination and repair. With the objective of understanding how IE1 is brought to PML, we discovered that PML aids the SUMOylation of IE1. This finding led us to identify a putative SUMO interaction motif on IE1 that

**Funding:** This work was funded by a Canadian Institutes of Health Research grants (www.cihr-irsc.gc.ca) (MOP_123214 and PJT_156118) awarded to LF and an European Research Council (ERC) (https://erc.europa.eu) grant (Stg 677673) awarded to BBK. The funders had no role in study design, data collection and analysis, decision to publish, or preparation of the manuscript.

**Competing interests:** The authors have declared that no competing interests exist.

is essentials for both its SUMOylation and IE1 oligomerization with PML-NBs. We next studied the role of PML on HHV-6B integration and identified that cells that are deficient for PML were less susceptible to HHV-6B integration. These results correlate with the fact that PML influences IE1 localization at telomeres, the site of HHV-6B integration. Our study further contributes to our understanding of the mechanisms leading to HHV-6B chromosomal integration.

## Introduction

Human herpesvirus 6B (HHV-6B) is a *betaherpesvirus* infecting nearly 90% of the population worldwide. HHV-6B is the etiologic agent of *exanthem subitum*, a childhood disease whose symptoms include fever, occasional skin rash and respiratory distress [1]. Following primary infection, HHV-6B enters in latency. In immunocompromised patients such as hematopoietic stem cell transplantation recipients, HHV-6B frequently reactivates from latency and can cause serious medical complications [2–4]. During latency, most herpesviruses maintain their genome as a circularized episome. Episomes of some herpesviruses (EBV and KSHV) can be tethered to human chromosomes, ensuring the transfer of the virus genome to both daughter cells following cell division [5,6]. Thus far, the state of the HHV-6B genome during latency remains elusive.

Importantly, HHV-6B can readily integrate its genome into host chromosomes [7–9]. HHV-6B integration can take place in various chromosomes but invariably occurs within the telomeric region [10–12]. Telomeres are non-coding $(TTAGGG)_n$ hexanucleotides present at the chromosome termini. They protect chromosomes against the loss of genetic information by preventing the recognition of chromosome ends by the DNA damage response (DDR) machinery. Moreover, they serve as a buffer zone to prevent premature cell senescence due to the end replication problem of the cellular DNA polymerase. Interestingly, HHV-6B can integrate into the chromosomes of germinal cells and upon fertilization, the resulting embryo will carry a copy of the viral genome in every cell [8]. The resulting individual will transmit the integrated HHV-6B to half of its descendants. Such individual harbors one or more integrated copy of HHV-6B per cell and are referred to as inherited chromosomally-integrated HHV-6B (iciHHV-6B) subjects [3,13]. Large-scale studies have identified iciHHV-6 as a predisposing factor for angina pectoris as well as pre-eclampsia in pregnant women [14,15]. Viral integration into telomeres has been suggested to be an alternative mechanism to maintain the HHV-6B genome during latency [11] as reported for the highly oncogenic Marek's disease virus [16]. In order for an integrated viral genome to reactivate, replicate and form new virions, the integrated genome must remain whole. In support, the integrated HHV-6B genome is generally intact and conserved without any gross rearrangements or mutations [17]. Furthermore, the integrated HHV-6B genome can express genes and lead to complete viral reactivation [2,10,18,19]. Until now, viral or cellular proteins involved in HHV-6B integration remains to be fully characterized. In a recent report, we have provided evidence that the telomeric shelterin protein TRF2 is required for efficient integration of HHV-6A and HHV-6B [20].

An interesting candidate that is potentially involved in HHV-6B viral integration is the immediate-early (IE) 1 protein (IE1) [21–23]. IE1 regulates early (E) gene expression and plays important roles in the replication of the virus during the lytic phase. Moreover, it establishes a favorable environment by manipulating PML-Nuclear bodies (PML-NBs) which are part of the cellular antiviral defense [24]. In the context of a viral infection, PML-NBs have been shown to repress replication of several viruses in collaboration with SP-100 and DAXX.

PML-NBs are found mostly in the nucleus and contain large quantities of the PML protein [25,26]. Many herpesviruses have established ways to overcome this antiviral mechanism by degrading or manipulating PML-NBs. For instance, herpes simplex virus 1 (HSV-1) encodes the E3 ubiquitin ligase ICP0 that conjugates ubiquitin to PML and induces its degradation [27,28]. The IE1 of human cytomegalovirus (hCMV) de-SUMOylates PML-NBs resulting in PML redistribution and also inhibits PML *de novo* SUMOylation [29]. PML-NBs are also involved in DNA damage repair. Marchesini et al. demonstrated that PML is essential for telomere maintenance in non-neoplastic cells, as cells undergo apoptosis in absence of PML after DNA damage at these sites [30]. Intriguingly, IE1 has been shown to colocalize with PML during HHV-6B infection without inducing the dispersal of PML-NBs [21–23,31]. However, the biological relevance of this PML-IE1 interaction remains unknown.

Considering that 1) PML localizes at telomeres, 2) PML-NBs associate with DNA repair proteins and 3) viral integration occurs at telomeres, we hypothesized that PML plays a role in HHV-6B chromosomal integration. Here we show that PML-NBs are an important site for multiSUMOylation of IE1. We also identified motifs providing that multiSUMOylation state proven to be essential for its nucleation and association with PML-NBs. We also demonstrate that IE1 not only localizes with PML, but also with the host telomeres. Lastly, we provide evidence that PML plays a role in HHV-6B chromosomal integration.

## Results

### IE1 associates with PML

Immediate early proteins of herpesviruses are the first expressed upon infection. We and Stanton et al. [23] have previously shown that IE1 of HHV-6B is found to localize with PML throughout infection, without destroying these nuclear bodies [20]. In this paper we found that the number of IE1 foci colocalizing with PML increases as infection progresses, as indicated by a higher Mander's coefficient of colocalization (MCC) at 72h of infection, compared to 24h (Fig 1A). Moreover, by comparing the volume of PML foci in non-infected (NI) relative to HHV-6B infected MOLT-3 cells (Fig 1B) we conclude that the mean volume of PML foci is enhanced after 72h of infection. Mean IE1 foci volume also increased as infection progressed (Fig 1B). Hence, the presence of HHV-6B does not disrupt or degrade PML-NBs formation but likely cause their fusion, in agreement with our previous results [21]. These experiments were performed in T lymphoblastoid cells fully permissive to HHV-6B lytic infection. We have previously shown that U2OS and HeLa cells are semi-permissive to infection with HHV-6B DNA replication occurring in a minority of cells, despite considerable expression of IE and E proteins [32]. Both cell lines have been extensively used by our group to assess HHV-6B integration [20,33–35]. We studied whether ectopically expressed IE1 would be found associated with PML in these cells and in the absence of other viral proteins (Fig 1C). 3D reconstitutions of deconvoluted immunofluorescence (IF) images (right column) show that IE1 mostly colocalizes with PML (represented in yellow color). As control, we used the IE2 protein that also displays a punctate nuclear distribution. However, the fact that the only anti-IE2 antibody available is specific for HHV-6A IE2 [36] prevented us from using HHV-6B IE2 as control. Unlike IE1, the majority of IE2 did not colocalize with PML. A scatter plot graph quantification for the MCC of deconvoluted acquisitions indicates that IE1 colocalizes almost perfectly with PML compared to IE2A (Fig 1D). Considering that PML is a family of proteins (n = 7) sharing common $NH_2$ terminals with varying C-terminal lengths, we studied the colocalization of IE1 with PML I-VI nuclear isoforms. IE1 was found to colocalize with all PML nuclear isoforms (S1 Fig) suggesting that IE1 recognizes a motif within a region of the $NH_2$ terminus common to all PML isoforms.

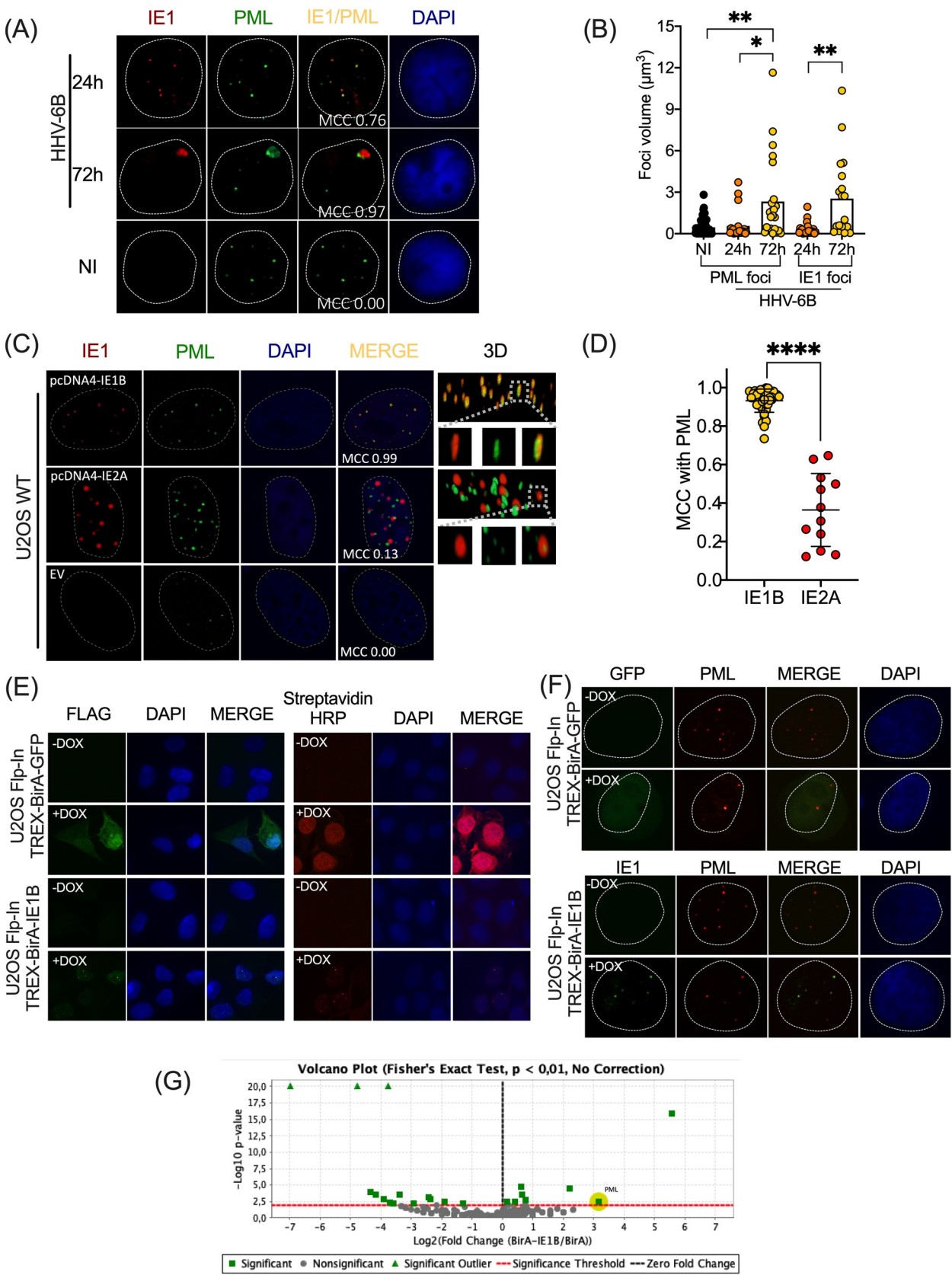

**Fig 1. IE1 associates with PML during infection and in the absence of other HHV-6B proteins.** (A) Confocal microscopy of IF images representing PML-NBs in non-infected (NI) MOLT-3 cells and infected with HHV-6B for 24 and 72h, acquired at 40X. Colocalization is presented for each time point. PML was detected using anti-PML with an anti-mouse-ALEXA-488 antibodies (green) and infected cells were detected with IE1 expression using anti-IE1-ALEXA-568 (red). (B) Graph showing the volume of PML and IE1 foci during infection. *P<0.01; **P<0.006, Kruskal-Wallis test (One-Way ANOVA with multiple comparisons). (C) Confocal microscopy of IF images representing ectopically expressed IE1 colocalizing with endogenous PML in U2OS cells, acquired at 63X. Cells were transfected with pcDNA4/TO-IE1B, pcDNA4TO-IE2A or pcDNA4/TO (EV: Empty Vector) vectors for 48 hours. IE1 and PML were detected with the same antibodies used in Fig 1A. IE2 was detected using anti-IE2-ALEXA-568 (red). 3D reconstitutions of deconvoluted acquisitions show perfect colocalization of IE1 with PML, represented by the yellow color. (D) Graph of MCC of IE1 (n = 39) and IE2A (n = 12) foci colocalizing with PML. Each dot represents a nucleus. ****P<0.0001, Mann-Whitney test. (E) U2OS-Flp-In TREX cells were co-transfected with pcDNA5/TO expression vectors containing FLAG-BirA-GFP or FLAG-BirA-IE1B with pOG44 (flipase) and selected with hygromycin (250µg/ml) and blasticidin (50µg/ml). IF confirms BirA-GFP and BirA-IE1B expression (Flag) and biotinylation of proteins (Streptavidin-HRP-594) when induced with doxycycline. (F) Stable U2OS-BirA-GFP and U2OS-BirA-IE1 cells were induced or not with doxycycline and colocalization of IE1 with PML is demonstrated following PML and FLAG labelling. (G) Volcano plot of IE1 specific interacting partners following mass spectrometry and a Fisher's Exact test to identify specific IE1 interacting partners. PML was one of the specific interacting IE1 partner, compared to BirA alone, with a p value of P<0.0034.

Using co-immunoprecipitation assay, we were unsuccessful in pulling down IE1 in association with PML, suggesting either a weak or a lack of physical interaction between these two proteins. We therefore made use of the BioID system that identifies close proximity interactors within a 10 nm radius [37]. This system takes advantage of the fusion between a protein of interest with a biotin ligase, BirA. Following the addition of biotin, interactors are marked with biotin and they can be pull-down and analyzed by mass spectrometry (MS). Stable U2OS cell lines expressing doxycycline (Dox)-inducible BirA-GFP (as control) or BirA-IE1 were generated. Left panels of Fig 1E show BirA-GFP and BirA-IE1 expression after Dox treatment. Right panels of Fig 1E show biotin labeling of proteins following Dox treatment. In BirA-GFP expressing cells, biotinylation was observed throughout the cell, following the cellular distribution of BirA-GFP. In BirA-IE1 expressing cells, biotinylation was restricted to a few foci, similar to the distribution of BirA-IE1. BirA-IE1 colocalized with PML, indicating that the fusion partner did not affect the ability of IE1 to localize with PML (Fig 1F). Using BioID, we could immunoprecipitate biotinylated PML in the presence of BirA-IE1, confirmed by the detection of PML by mass spectrometry (Fig 1G). Taken together, these results indicate that the majority of IE1 colocalizes with PML and it does so in the absence of any other viral factors.

## Expression of PML influences hyperSUMOylation of IE1

Next, we wanted to study how PML might recruit IE1 at PML-NBs and what effect it could have on IE1. PML is well-known for recruiting partners via SUMO-SUMO-Interacting Motif (SIM) interactions. PML-NBs are also a hotspot for SUMO modifications as proteins implicated in the SUMOylation process are found at PML-NBs [38–41]. We have previously shown that IE1 of HHV-6B is SUMOylated on lysine 802 (K802) [22]. To expand on these results, we have studied SUMOylation of IE1 in HEK293T cells overexpressing PML-I and HA-SUMO-1. HEK293T cells express low to no levels of PML and are easily transfectable [42]. Immunoprecipitation of IE1 followed by western blot for HA-SUMO-1 allowed us to detect HA-SUMO conjugated IE1 (Fig 2A). Expression of PML-I resulted in hyperSUMOylation of IE1, defined by a second and higher molecular weight SUMO-1-IE1 protein.

A given protein can display different SUMOylation states (mono, poly or multiSUMOylation) depending on the absence or presence of E3 SUMO ligases, the numbers of SUMO acceptor sites, or the SUMO paralogues conjugated (Fig 2B). SUMO branching is a state of SUMOylation that is SUMO paralogue dependent. In this regard, there are three major SUMO protein paralogues constitutively expressed in eukaryotic cells, SUMO-1 and SUMO-2/3, that can be attached to a SUMO acceptor site on a target protein. Unlike SUMO-1, SUMO-2/3 possess SUMO acceptor sites within their primary amino acid sequence, allowing

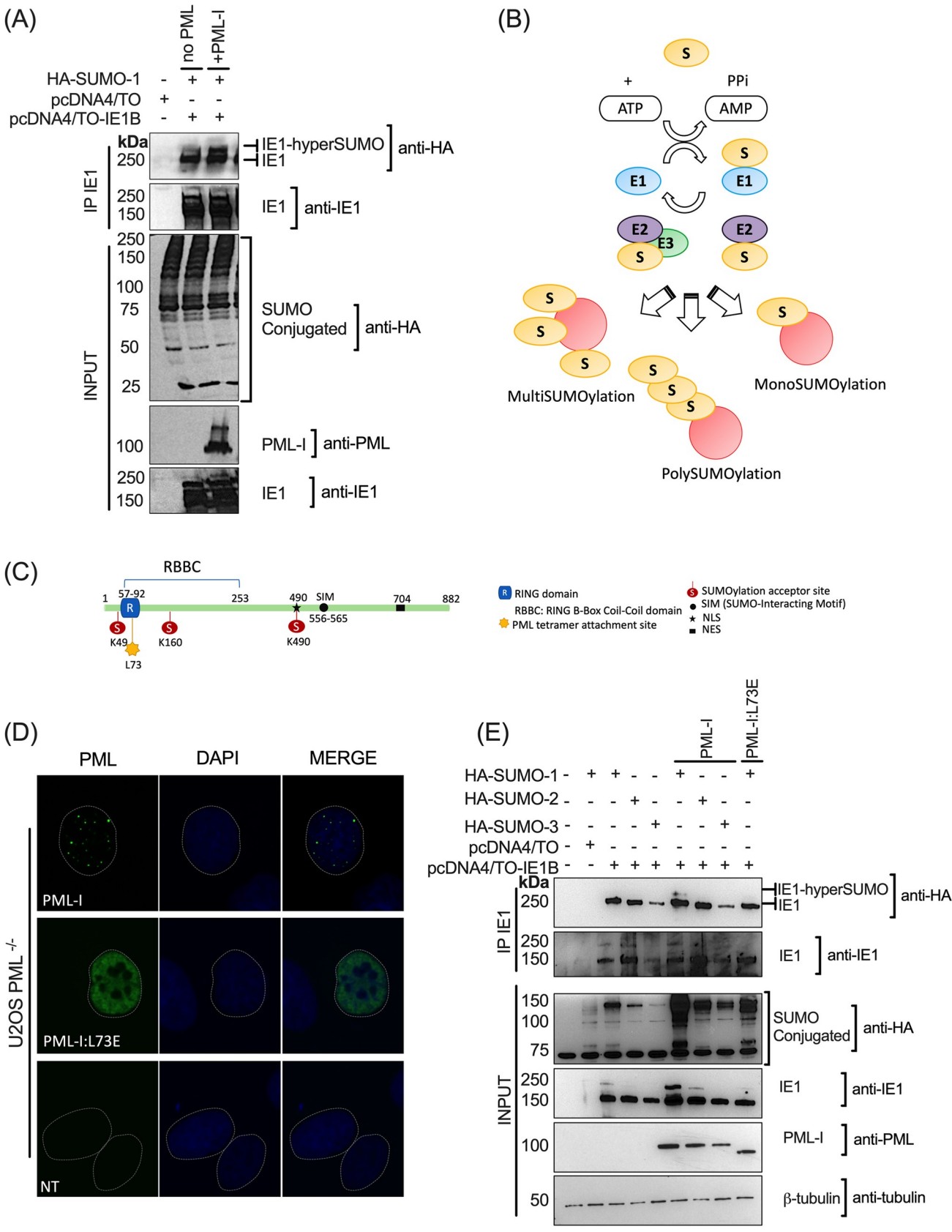

**Fig 2. Presence of PML-NBs leads to hyperSUMOylation of IE1.** (A) HEK293T cells were co-transfected at a ratio of 1:1:1 with pCMV2N3T-HA-SUMO-1: pDEST-MYC-PML-I: pcDNA4/TO or pcDNA4/TO-IE1B plasmids. After 48h cells were collected and lysed in RIPA buffer. A portion (10%) of the lysate was used to monitor protein expression (input). The remaining portion of the lysate was used for IE1 immunoprecipitation. IE1-SUMO-1 conjugates were detected using anti-HA antibodies. In presence of PML, SUMOylation of IE1 is enhance and that is shown by the presence of a second IE1-SUMO-1 band. (B) Illustration representing possible SUMOylation states of protein. Once mature, a SUMO protein is charged on a E1 activating enzyme and then transferred to a E2 conjugating enzyme. The SUMO-E2 complex can directly SUMOylate a protein on its acceptor lysine (K). Following conjugation to E2, the SUMO-E2 complex can be bound by a E3 SUMO ligase that will facilitate the discharge of the SUMO protein on the acceptor site of a target. The presence of a E3 SUMO ligase can influence or help reach those different levels of SUMOylation where polySUMOylation is realized by the presence of SUMO-2/3 paralogues that possess SUMO acceptor sites. (C) Scheme illustrating major domains and modification sites of PML protein. Leucine at position 73 allows the attachment of monomers of PML to form a tetramer, essential for the formation of PML-NBs [44]. (D) IF images representing the different PML-I phenotypes after transfection of PML-I WT or PML-I RING domain mutant (PML-I:L73E) in U2OS PML[-/-] cells. PML RING mutant lost the ability to form nuclear bodies. (E) HEK293T cells were co-transfected at a ratio of 1:1:1 with pCMV2N3T-HA-SUMO-1/SUMO-2 or SUMO-3: pCMV-MYC-PML-I: pcDNA4/TO or pcDNA4/TO-IE1B plasmids. Cells were collected and processed as in (A). Enhancement of SUMOylation of IE1 is PML-I WT and SUMO-1 dependent. These results are the representation of three independent experiments (n = 3).

SUMO-SUMO branching, resulting in polySUMOylated proteins [22,43]. To evaluate whether the formation of PML-NBs is directly responsible for IE1 SUMOylation, we generated a PML protein that can no longer form PML-NBs. We introduced a L73E mutation in the RING domain of PML-I (PML-I:L73E), a mutation proven to be essential for the tetramerization of PML monomers to form PML-NBs (Fig 2C) [44]. Compared to endogenous PML and ectopically expressed WT PML-I, PML-I:L73E has a diffuse nuclear pattern that does not allow the formation of PML-NBs (Fig 2D). We next determined if hyperSUMOylation of IE1 observed in Fig 2A was the results of SUMO-branching. Upon overexpression of WT PML-I, SUMO-IE1 was detected with all three SUMO paralogues (Fig 2E). Also of interest, hyperSUMOylated IE1 was only detected in the presence of WT PML-I and SUMO-1 but not with SUMO 2/3 or in the presence of PML-I:L73E. This suggests that hyperSUMOylation is not the result of SUMO-2/3 branching on IE1. Moreover, IE1 requires the presence of PML-NBs to be hyperSUMOylated. These findings argue for the existence of more than one SUMO acceptor sites on IE1. Hence hereforward, hyperSUMOylation of IE1 will be referred to as multiSUMOylation.

## MultiSUMOylation of IE1 is sequence motif and PML-NBs dependent

SUMOylation of a protein on its SUMO acceptor site can be influenced by the presence of a SUMO-Interaction Motif (SIM) that binds free SUMO or a SUMOylated protein in a non-covalent manner, allowing a change in the function or localization of a protein [45,46]. Using *in silico* analyses, we have identified putative SIM sites at position 573–575 (VVV) and position 775–777 (VIV) (Fig 3A and S3 Fig). To characterize the importance of these sequence motifs, we generated IE1 mutants by substituting the VVV or VIV sites by AAA, a mutation reported to abolish the binding of SUMO to SIM sites [47,48] (Fig 3A). We have also generated from these mutants a point mutation in which the known SUMO acceptor site, K802, was substituted to an arginine (K802R). Upon immunoprecipitation of WT IE1, we could detect robust SUMO-1 conjugation (Fig 3B). In contrast, SUMOylation of the $^{775}AAA^{777}$ mutant was much weaker while SUMOylation of the $^{573}AAA^{575}$ mutant was equivalent to that of WT IE1. Complete loss of IE1 SUMOylation was observed with the IE1$^{775}AAA^{777}$:K802R double mutant. Residual SUMOylation was observed with the IE1$^{573}AAA^{575}$:K802R mutant, arguing for the existence of an SUMO acceptor site other than the previously reported K802 [22]. IE1 with both putative SIM sites mutated, also displayed residual SUMOylation (Fig 3B). Taking into account the low level of endogenous PML expression in HEK293T cells [42] no multiSUMOylation of IE1 was detected in the absence of PML overexpression. The experiments were therefore repeated in HEK293T cells overexpressing PML-I (Fig 3C). In the presence of PML-I, multiSUMOylation of WT IE1 was observed and lost with the $^{775}AAA^{777}$ mutant, arguing in

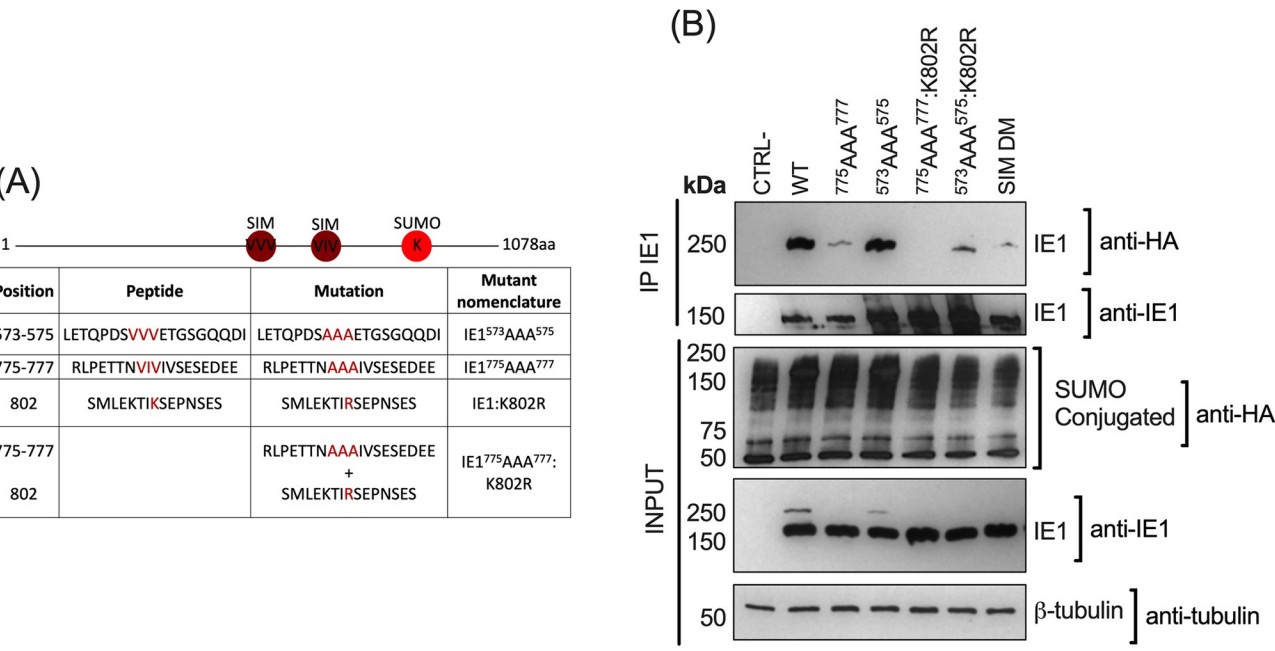

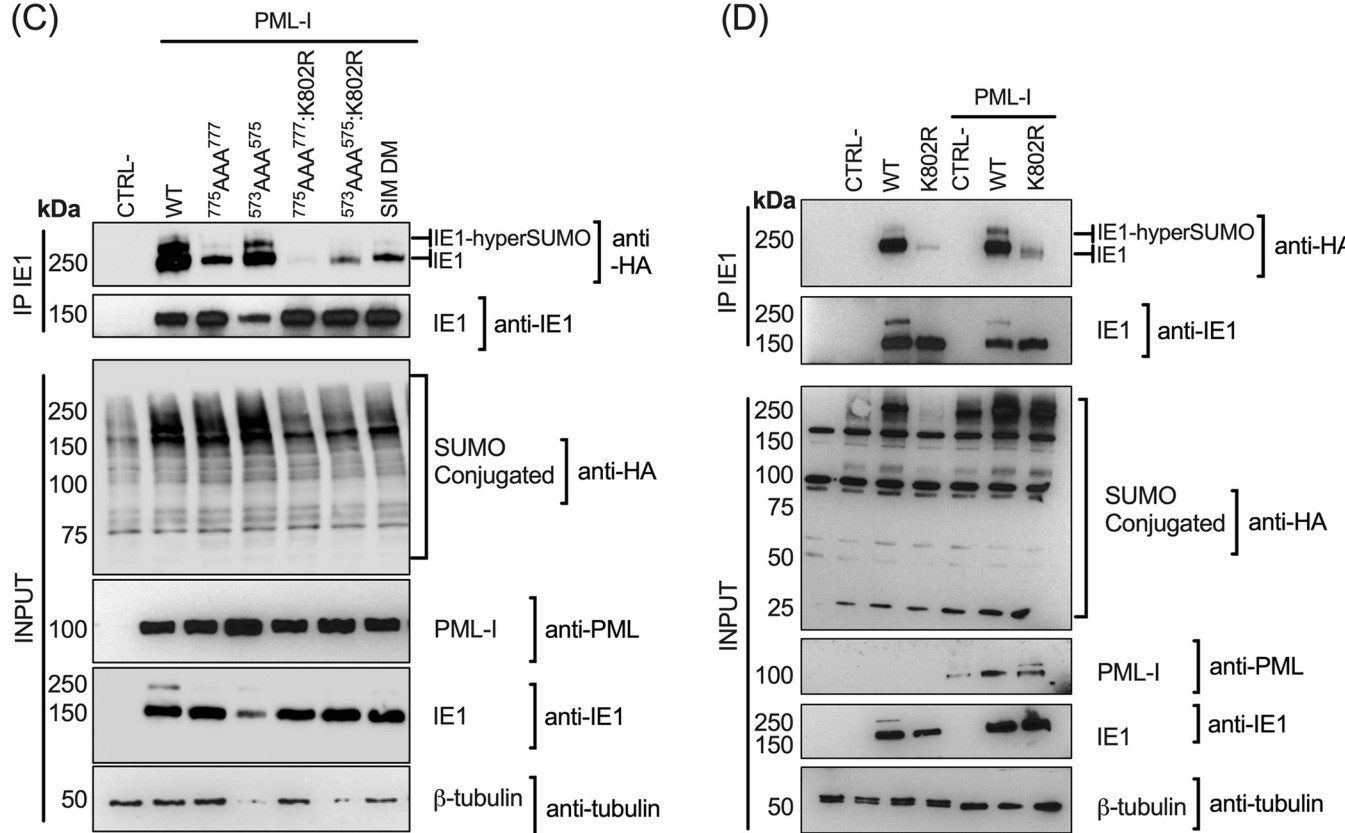

**Fig 3. Putative IE1 Sumo-Interacting Motif site is important for multiSUMOylation in presence of PML.** (A) Scheme and table of potential SUMO acceptor sites and SUMO-interacting motifs (SIM) of IE1 and the nomenclature of the generated mutants. (B) HEK293T cells were co-transfected with a ratio

of 1:1:1 of pCMV2N3T-HA-SUMO-1: pcDNA4/TO-IE1B WT, mutants or control: no plasmid, expression vectors. Cells were collected and processed as in Fig 2A. The sequence motif at position $^{775}$AAA$^{777}$ and the lysine at position 802 are essential for SUMOylation of IE1. (C) HEK293T cells were co-transfected with a ratio of 1:1:1 of pCMV2N3T-HA-SUMO-1: pcDNA4/TO-IE1B WT, mutants or control: PML, expression vectors. Cells were collected and processed as in Fig 2A. The sequence motif at position $^{775}$AAA$^{777}$ and the lysine at position 802 are essential for higher SUMOylation state of IE1. (D) Western blot of WT and K802R SUMOylation of IE1 in HEK293T overexpressing or not PML-I. Cells were collected and processed as in Fig 2A.

favor of a role for the VIV sequence motif in the multiSUMOylation process of IE1 by PML-I. Consistent with results of Fig 3C, mutation of the $^{573}$VVV$^{575}$ had no effect on the multiSU-MOylation of IE1 in presence of overexpressed PML-I. Overall, multiSUMOylation of IE1 in the presence of PML-I is greatly affected by mutation of the putative $^{775}$VIV$^{777}$ SIM site. More-over, and consistent with results of Fig 3B, IE1 SUMOylation was lost with the $^{775}$AAA$^{777}$: K802R double mutant. Of interest, SUMOylation of IE1 could be detected with the K802R mutant, indicating the presence of an additional SUMO acceptor site on IE1 (Fig 3D). Taken together, our results indicate that in the presence of PML-NBs, multiSUMOylation of IE1 is observed and is dependent on the presence of the putative $^{775}$AAA$^{777}$ SIM site.

## MultiSUMOylated IE1 contributes to nuclear bodies accumulation

Until now, it was unknown how IE1 manages to agglomerate into distinct foci within the nucleus. Thereupon, we next studied the phenotypes of IE1 mutants. Fig 4A shows that WT IE1 forms dot-like patterns (punctate) in the nucleus. However, when analyzing the phenotype of IE1:$^{775}$AAA$^{777}$:K802R, we observed two types of patterns. One pattern had small dot-like foci with a slightly diffuse signal. The other pattern was a diffuse IE1 nuclear distribution. Quantification of these patterns in U2OS cells indicates that IE1:$^{775}$AAA$^{777}$:K802R forms less IE1 foci compared to WT (Fig 4B). This indicates that SUMOylation of IE1 is important for IE1 oligomerization. We next wanted to look at the colocalization of WT IE1 and IE1:$^{775}$AAA$^{777}$:K802R with PML. Because the signal of IE1:$^{775}$AAA$^{777}$:K802R foci were less abundant and diffused, we could not determine the MCC. From a qualitative point of view, Fig 4C shows in 3D that the colocalization of IE1 mutant is similar to the WT, after deconvolution and subtracting the diffuse pattern. We next looked at the ability of IE1:$^{775}$AAA$^{777}$:K802R to form nuclear bodies (NB) with PML by comparing the volume of PML and IE1 foci (Fig 4D). Presence of WT IE1 resulted in greater PML foci volumes. In the presence of IE1:$^{775}$AAA$^{777}$: K802R the mean volume of the PML foci was comparable to that of the empty vector (EV) condition. Similarly, the mean volume of the IE1 foci was much larger for WT IE1 relative to the IE1:$^{775}$AAA$^{777}$:K802R mutant. Taken together, the presence of PML-NBs causes multiSU-MOylation of IE1 which could be responsible for the formation of a protein network, resulting in larger NBs at PML-NBs (Fig 4E).

## PML is required for efficient HHV-6B chromosomal integration

Considering that IE1 colocalizes with PML, is multiSUMOylated in the presence of PML-NBs and that a significant proportion of PML-NBs are present at telomeres (S4 Fig), the site of HHV-6B integration, we wanted to study the role of PML in HHV-6B integration. We gener-ated PML knockout (KO) U2OS cells using the CRISPR-Cas9 technology involving guide RNAs targeting a sequence in exon 1 of PML (S5 Fig). Following transfection and selection of cells, the deletion of a part of exon 1 resulting in a pre-mature STOP codon was confirmed by sequencing (S5 Fig).

U2OS PML$^{+/+}$ and PML$^{-/-}$ cell lines (Fig 5A) were infected with HHV-6B and integration frequencies were assessed four weeks post infection by droplet digital PCR, as described previ-ously [33] (S6 Fig). The percentage of integrated HHV-6B determined following ddPCR

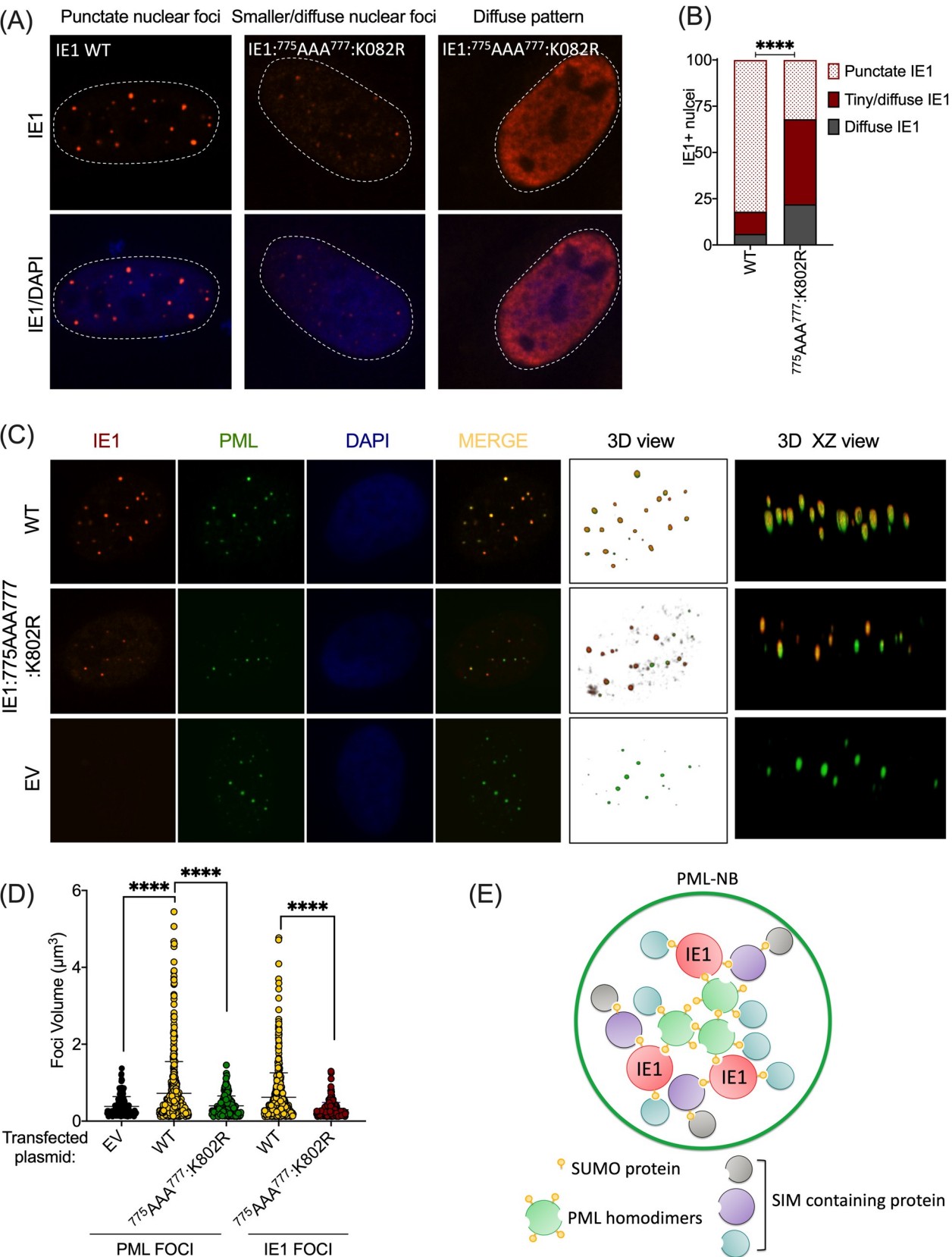

**Fig 4. MultiSUMOylated IE1 contributes to nuclear bodies accumulation.** (A) Confocal microscopy showing IF images of patterns of IE1 in U2OS transfected with WT IE1 and $^{775}$AAA$^{777}$:K802R mutant. (B) Graph showing the quantification of pattern in U2OS between WT IE1 (n = 85) and $^{775}$AAA$^{777}$:K802R mutant (n = 32). Total IE1$^+$ nuclei were counted and divided in three categories: punctate, tiny dots/diffuse and totally diffuse patterns. Data were normalized on 100. $^{****}$P<0.0001, chi-square. (C) Deconvoluted IF of IE1 WT and $^{775}$AAA$^{777}$:K802R mutant, 48 hours post-transfection. (D) Graph representing the volume of each PML and IE1 foci per nucleus with EV (n = 24) WT IE1 (n = 45) and $^{775}$AAA$^{777}$:K802R mutant (n = 20) $^{****}$P<0.0001, chi-square. (E) Hypothesis of interacting network of IE1 following multiSUMOylation by PML. MultiSUMOylation of IE1 in PML-NBs allow the recruitment of SIM-containing interacting partners at the additional SUMOylated site of IE1, resulting in bigger nuclear bodies.

analysis indicated that the HHV-6B integration frequency was significantly reduced in both U2OS PML$^{-/-}$ cell clones tested (Table 1). Next, we addressed whether the observed reduction in integration frequencies could be a consequence of increased lytic viral replication in PML$^{-/-}$ cells, as it was recently described for lymphoblast cells infected with HHV-6A [31]. Results indicate that there is no difference in the number of HHV-6B genome copies between U2OS PML$^{+/+}$ and PML$^{-/-}$ cells 24h post infection (Fig 5B).

HeLa LT PML$^{-/-}$ cells were also generated using the same procedure used for the U2OS cells. As additional control, PML expression was restored in HeLa LT PML$^{-/-}$ clone #1 (Fig 5C). This was achieved by transducing the PML$^{-/-}$ cells with the lentivirus PML-I. In the absence of PML, HHV-6B integration frequencies were lower in both clones tested compared to HeLa LT PML$^{+/+}$ cells and recovered upon restoration of PML (Table 2). As with U2OS cells, there was no difference in the HHV-6B genome copy numbers between HeLa LT PML$^{+/+}$ and PML$^{-/-}$ cells 24h post infection, suggesting that the absence of PML does not enhance HHV-6B ability to replicate in semi-permissive U2OS and HeLa LT cells (Fig 5D). Collectively, our data demonstrate that PML is required for efficient HHV-6B integration in two distinct cell types.

## IE1 colocalizes at telomeres during HHV-6B infection

It is known that a majority of PML-NBs can be found at telomeres of Alternative Lengthening of Telomeres+ (ALT+) cell lines such as U2OS cells [30,39]. This PML-NBs/telomere complex is called APBs (ALT-PML-Nuclear Bodies) and is a hallmark of ALT+ cell lines in which significant DNA repair occurs to allow telomere elongation through homologous recombination events [49,50]. Considering that PML is being often found at telomeres, that telomeres are the preferred sites of HHV-6B integration, and that PML deficient cells integrated HHV-6B less frequently, we were interested in knowing whether IE1 would localize at telomeres during infection in a cell line used to study integration. To answer this question, U2OS cells were infected with HHV-6B for 48 hours and analyzed for IE1 expression and localization by confocal microscopy. IE1 was detected as distinct nuclear foci during infection (Fig 6A) with a proportion of IE1 colocalizing with telomeres (yellow square). Following the deconvolution of each acquisition for each channel, 3D quantification of z stacks revealed that 31.3%±19.7 of IE1 foci colocalized with telomeres (Fig 6B). See also S2 Fig for IF-FISH example of colocalization in the z stack. Additionally, IE1 colocalized with PML at telomeres of ALT$^-$ cells such as HeLa LT cells (telomerase+) (S7 Fig). We made use of HeLa LT cells (HeLa cells with long telomeres) to be able to visualize and quantify the colocalization events. This indicates that the associations of IE1/PML at telomeres occurs in both ALT+ (U2OS) and telomerase expressing cells (HeLa) cells used for *in vitro* HHV-6B integration assays.

## IE1 localization at telomeres is influenced by the presence of PML

Considering that the presence of PML influences the level of HHV-6B integration and that IE1 colocalizes at telomeres, we asked whether the colocalization of IE1 at telomeres would be affected by the absence of PML in U2OS cells. Following transfection of IE1 expression vector

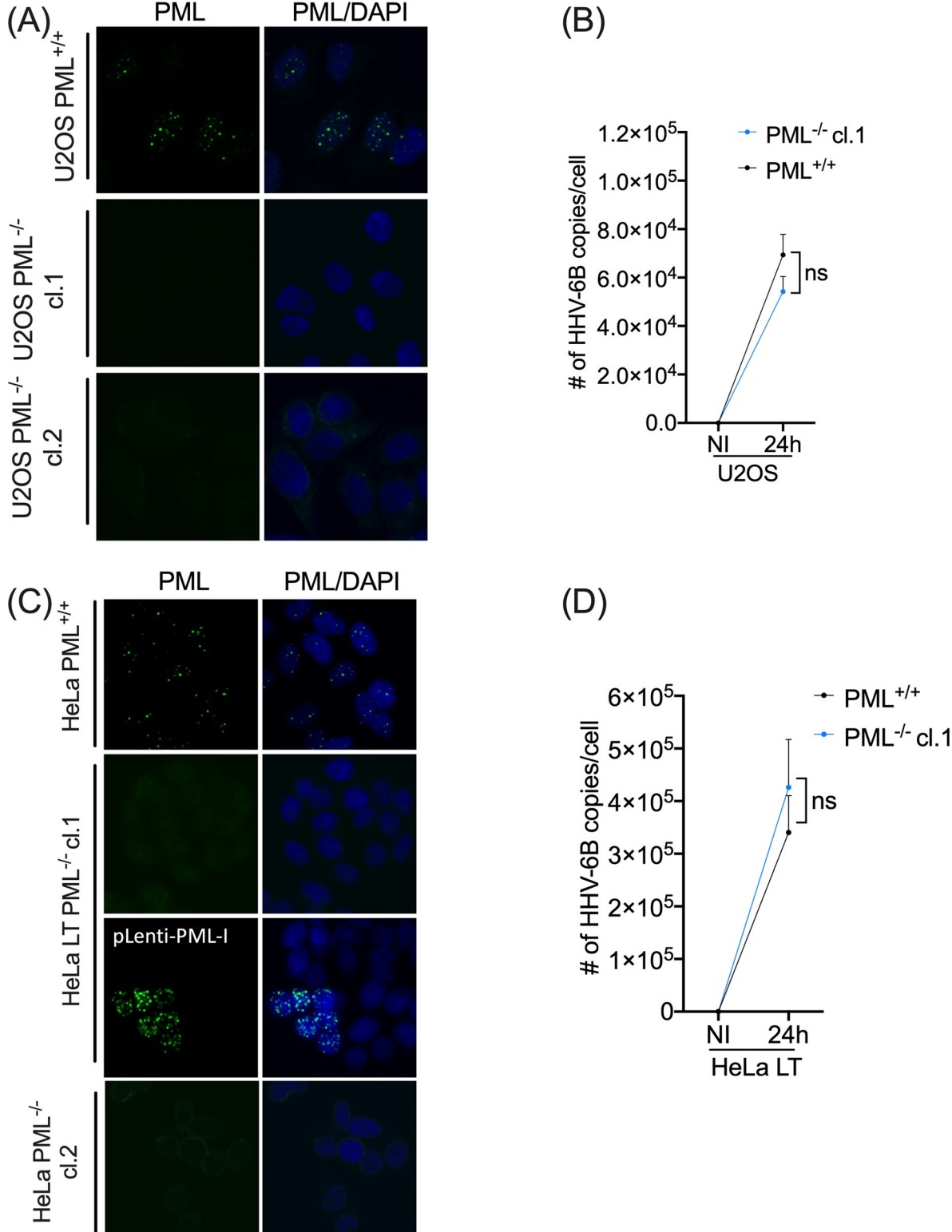

**Fig 5. Absence of PML affects HHV-6B integration efficiency.** (A) IF images of PML expression in WT and PML$^{-/-}$ U2OS cells. PML is shown in green. (B) PML$^{+/+}$ and PML$^{-/-}$ cells were infected at a MOI of 1 with HHV-6B and were collected at day 0 and 24 hours and cellular DNA was extracted. A qPCR was done to determine the number of copies of HHV-6B by using primers for *U65-66* gene for HHV-6B and *RPP30* as a cellular reference gene. Data presented are the mean of three independent experiment and were analyzed using the Mann-Whitney test. P>0.05 (not significant). (C) IF images of PML expression in WT, PML$^{-/-}$ and PML$^{-/-}$—PML-I HeLa LT cells. PML is

shown in green. Restored expression of PML was done in HeLa PML$^{-/-}$ clone 1. HeLa PML$^{-/-}$ clone 1 cells were transduced with a lentivirus expressing PML-I (pLenti-PML-I). After hygromycin selection, HeLa LT PML$^{+/+}$, HeLa LT PML$^{-/-}$, HeLa LT PML$^{-/-}$clone 1-PML restored were used for IF and for integration assay. (D) HeLa LT PML$^{+/+}$,PML$^{-/-}$ and HeLa LT PML$^{-/-}$clone 1-PML restored were infected at a MOI of 1 with HHV-6B. Cells were processed as in Fig 5B. P>0.05 (not significant).

in U2OS PML$^{+/+}$ and PML$^{-/-}$ cells, deconvoluted IF acquisitions showed that IE1 localization at telomeres was reduced in the absence of PML (Fig 6C and 6D). The mean MCC of IE1 at telomeres was reduced from 0.36 to 0.12 (Fig 6D) (***p<0.0001). We also quantified whether the number of IE1 nuclear foci differed between the cell lines as a possible explanation for the observed lower frequency of IE1 at telomeres in U2OS PML$^{-/-}$. Results in Fig 6E show that there are no significant changes in the number of IE1 foci between PML$^{+/+}$ and PML$^{-/-}$ cells. Altogether, this indicates that IE1 localizes at a lower frequency at telomeres in the absence of PML, correlating with the results of the HHV-6B integration assay.

SUMOylation of IE1 was also studied in cells used for HHV-6B integration. U2OS PML$^{+/+}$ and PML$^{-/-}$ cells were co-transfected with pCMV2N3T-HA-SUMO-1 and pcDNA4-MY-C-IE1B. Immunoprecipitation of IE1 showed that multiSUMOylation of IE1, characterized by a second and higher molecular weight band, was only present in U2OS PML$^{+/+}$ (Fig 6F). Altogether, the presence of PML-NBs positively influences the multiSUMOylation of IE1 and its presence at telomeres.

## Discussion

In the current study we showed that: 1) IE1 colocalizes with all six PML nuclear isoforms, independently of other viral factors; 2) in the presence of PML, IE1 is multiSUMOylated; 3) the formation of PML-NBs is required for IE1 multiSUMOylation; 4) the IE1 $^{775}$VIV$^{777}$ putative SIM site is required for efficient IE1 SUMOylation and multiSUMOylation; 5) IE1 association with PML nuclear bodies is dependent on the $^{775}$VIV$^{777}$ site and K802; 6) PML is required for optimal HHV-6B integration in the telomeres of host chromosomes and 7) IE1 localization at telomeres is partly dependent on the presence of PML.

One major interest in the field is to identify proteins that participate in the HHV-6B chromosomal integration. One hypothesis initially raised was that HHV-6B chromosomal integration would require telomerase, the enzyme responsible for telomere elongation [51]. However, telomerase proved not essential as we have shown that HHV-6B integration occurs in both telomerase negative and positive cells [33,34]. Cellular telomeres are protected by a protein complex called shelterin, whose main function is to protect chromosome ends from being recognized as damaged DNA by DNA repair proteins. In recent work, we have demonstrated that TRF2 binds to HHV-6A/B telomeric repeats and that reducing the levels of TRF2, lower the rates of HHV-6A/B integration [20]. We have argued that TRF2 may serve to protect the viral DNA ends from DDR and maintain the viral genome integrity.

Table 1. Importance of PML for HHV-6B chromosomal integration in U2OS cells.

| U2OS | % of cells with integrated HHV-6B * (n) ** | p value |
|---|---|---|
| U2OS PML $^{+/+}$ | 1.96±0.0570 (9450) | |
| U2OS PML $^{-/-}$ clone 1 | 0.86±0.2273 (33320) | <0.0001 |
| U2OS PML $^{-/-}$ clone 2 | 0.90±0.08639 (33110) | <0.0001 |

* mean ± SD of three independent cultures

** total number of cells analyzed

**Table 2. Importance of PML for HHV-6B chromosomal integration in HeLa LT cells.**

| HeLa | % of cells integrated HHV-6B * (n) ** | p value |
|---|---|---|
| HeLa PML $^{+/+}$ | 2.66±0.8822 (32790) | |
| HeLa PML $^{-/-}$ clone 1 | 1.23±0.5894 (43030) | <0.0001 |
| HeLa PML $^{-/-}$ clone 2 | 1.86±0.0796 (14480) | <0.0001 |
| HeLa PML $^{-/-}$ clone 1 +PML restored | 2.63±0.5729 (30516) | ns |

* mean ± SD of three independent cultures

** total number of cells analyzed

Proteins other than shelterin are localized at telomeres. In telomerase negative cells such as U2OS cells, telomeres are mainly elongated by the presence of APBs [39,52,53]. These nuclear bodies are primarily formed by the PML protein itself that recruits hundreds of interacting partners at telomeres such as helicases implicated in G-quadruplex structure resolution like the Bloom syndrome protein (BLM), the Werner Syndrome Protein (WRN) and other proteins implicated in DNA recombination and repair [30,39,54–56]. Osterwalds et al. [39] have shown that in ALT+ cells such as U2OS, PML-NBs (APBs) are frequently present at telomeres. We have confirmed this result (S4 Fig). We also noticed that a proportion of PML-NBs localized to telomeres of telomerase expressing cells such as HeLa LT cell (S4 Fig), in agreement with Marchesini et al [30]. These findings support the role for PML in the DNA repair mechanisms [30,49,57].

The HHV-6B genome is about 160 kilobase pairs (kbp) in length and contains a unique region (U) with close to 100 open reading frames [58–60]. This U region is flanked by identical direct repeat regions (DR$_L$ and DR$_R$) of 8–9 kbp that contain telomere arrays identical to human telomeres at both ends [59,61]. Because of the existing homology between HHV-6B terminal sequences and telomeres, integration could be the result of homologous recombination events. The observation that a mutant of HHV-6A lacking telomeric repeats integrates much less efficiently supports this hypothesis [62].

Telomeres are protected by the shelterin complex to prevent DNA damage recognition at telomeres and repair. One shelterin protein in particular, TRF2, blocks DNA damage sensor and repair proteins such as the Ataxia-telangiectasia-mutated (ATM) at telomeres, a pathway that senses double-stranded DNA breaks [63–65]. Interestingly, when PML-NBs are present at telomeres, TRF2 is SUMOylated by MMS21, resulting in a lower density of TRF2 on telomeres, arguing that such site may be more prone to recombination events [66].

Here we reported that the IE1 protein is mainly found in association with PML. Among other purposes, PML-NBs serve as a hub for SUMO modifications. Briefly, the SUMOylation steps involve the activation of a mature SUMO protein by the activating enzyme E1(SAE1/2) in an ATP-dependent manner. Once SUMO is activated (terminal di-glycine motif) it is transferred to a E2 conjugating enzyme (Ubc9). SUMO can then be transferred to an acceptor lysine present within a SUMO consensus acceptor site (ΨKXE/D) on a target protein. This transfer results in an isopeptide bond of the terminal glycerin on SUMO. The transfer of a mature SUMO protein to a target protein can be aided by an E3 SUMO ligase that can directly bind the E2 enzyme or the targeted protein (Fig 2B) [67]. Considering that PML-NBs are a hub for SUMO modifications and that IE1 protein of HHV-6B can be SUMOylated [21,22] it was conceivable to us that the presence of PML might affect IE1 SUMOylation status. We demonstrated that not only the presence of PML-NBs enhances IE1 SUMOylation but also causes multiSUMOylation of IE1 (Fig 2). It is tempting to speculate that PML might act a SUMO ligase for HHV-6B IE1, as was initially reported for the CMV IE1 protein [42].

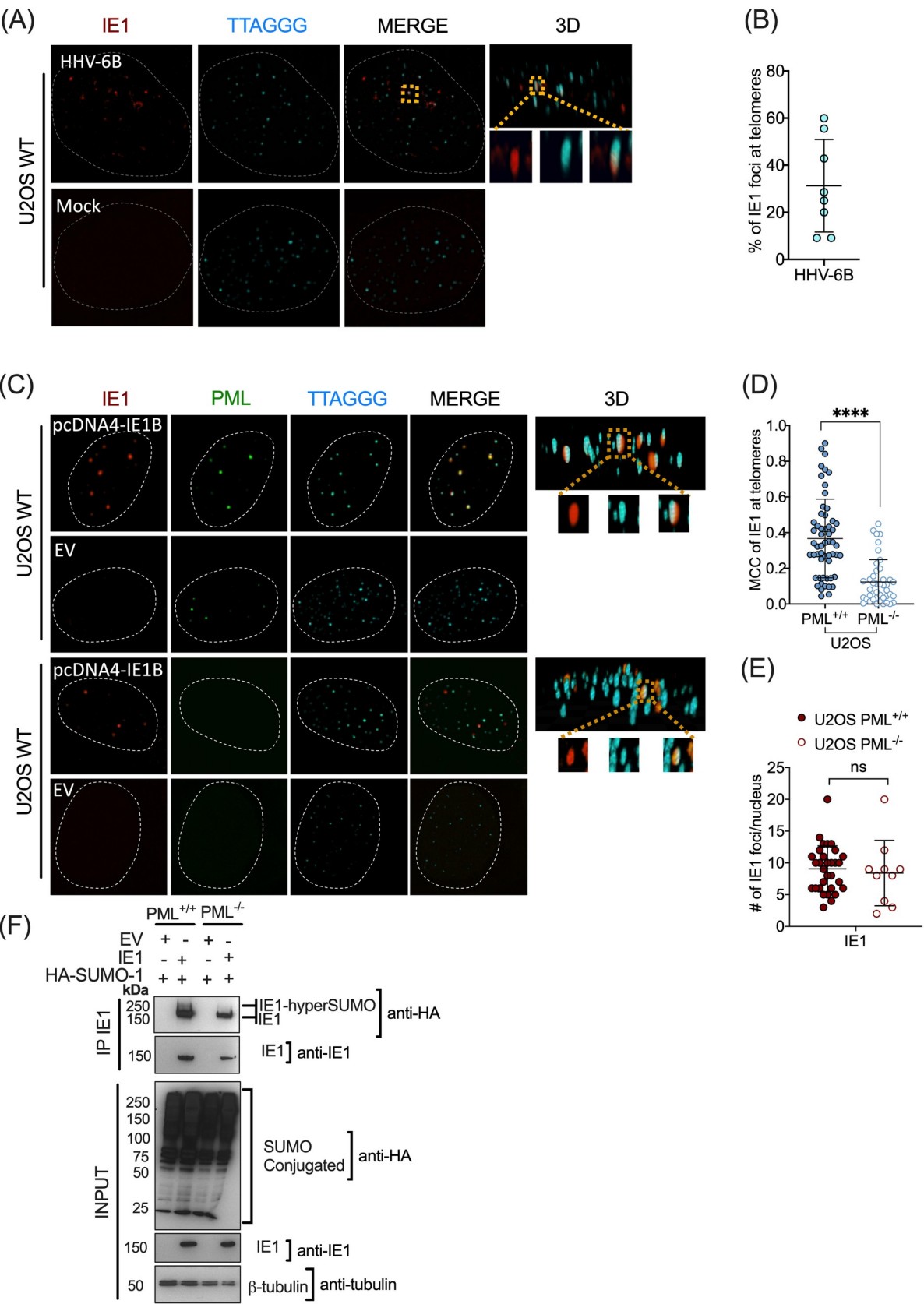

**Fig 6. Localization of IE1 at telomeres is influenced by the presence of PML.** (A) Deconvoluted confocal microscopy images representative of IE1/telomere staining of U2OS cells infected with HHV-6B for 48 hours. Telomeres were detected using a Cy5-labeled telomeric probe (aqua). Yellow squares represent IE1 colocalizing with telomeres in 3D. (B) Graph showing the percentage ± sd of IE1 (n = 8) foci colocalizing at telomeres throughout the z stacks, in HHV-6B infected U2OS cells. (C) Confocal microscopy images of U2OS PML$^{+/+}$ and $^{-/-}$ cells transfected with pcDNA4/TO-IE1B or empty vector (EV). Yellow squares represent IE1 colocalizing with telomeres. (D) Graph showing the proportion (mean±sd) of IE1 foci localizing at telomeres throughout the z stacks, in U2OS PML$^{+/+}$ (n = 46) and PML$^{-/-}$ (n = 42) cells. Each dot represents one IE1$^+$ nucleus. Data were transformed to log values and the normal distributions analyzed using an unpaired t test with Welch's correction to compare frequency of IE1 at telomeres. $^{****}P<0.0001$. (E) Graph showing the number of IE1 foci in PML$^{+/+}$ and PML$^{-/-}$ cell lines. P = ns, unpaired t test. (F) U2OS PML $^{+/+}$ and $^{-/-}$ cells were co-transfected at a ratio of 1:1:1 with pCMV2N3T-HA-SUMO-1: pcDNA4/TO or pcDNA4/TO-MYC-IE1B plasmids. After 48h cells were collected processed as in Fig 2A.

However, recent results challenge these findings and rather propose that CMV IE1 SUMOylation occurs at the nucleosomes possibly via the PIAS1 E3 SUMO ligase [68]. Analysis of HHV-6B IE1 primary sequence revealed no homology with the CMV IE1 chromatin tethering domain [69]. Whether HHV-6B IE1 associates with nucleosomes remains to be examined experimentally with more work needed before concluding on the SUMO ligase role of PML responsible for IE1 multiSUMOylation.

Previous results prompted us to determine whether IE1 possesses SIM sites that could allow IE1 to interact with SUMOylated proteins (other than PML) at telomeres. Studies done with Varicella-Zoster Virus ORF16 protein have shown that ORF16 SIM site is important for its role on disrupting PML-NBs [47]. Additionally, although Ubc9 can bind SUMO directly at a SIM site, the interaction is weak [70]. This can be strengthened with the help of E3 SUMO ligase [70]. Accordingly, we have demonstrated that the $^{775}$VIV$^{777}$ putative SIM site is essential for complete SUMOylation of IE1 and for its multiSUMOylation in presence of PML-NBs (Fig 3B and 3C). The double $^{775}$VIV$^{777}$:K802R mutant lost its ability to be SUMOylated indicating that both the VIV and K802 are important for IE1 SUMOylation. Our results also indicate that K802R could still be SUMOylated, although not multiSUMOylated (Fig 3D). Because PML can bind Ubc9 (E2 enzyme) [71], we propose that SUMOylated PML interacts with the putative IE1 SIM site bringing along a SUMO-1 charged E2 that adds one more SUMO-1 on IE1 to a yet to be identified acceptor residue.

Our work also demonstrated the importance SUMOylation of IE1 for its nuclear distribution as the $^{775}$VIV$^{777}$:K802R mutant fails to properly form punctate foci (Fig 4A and 4B). Moreover, SUMOylation sites of IE1 are important for the association of IE1 with PML (Fig 4C and 4D). Our results suggest that the multiSUMOylated state of IE1 enables the formation of bigger NBs which influences the interactome network of IE1 with a variety of other SUMOylated or SIM containing proteins (Fig 4E).

To further understand the role of PML in HHV-6B biology, we tested whether PML played a role in HHV-6B chromosomal integration. To do so, we used PML KO cell lines (Fig 5). For each cell line used, two independent PML KO clones were tested to ensure reproducibility and avoid potential CRISPR-mediated off-target effects. In U2OS cells (Table 1), HHV-6B integration was less frequent in PML$^{-/-}$ cells (p<0.0001). In HeLa LT cells (Table 2), the same effect was observed and was rescued when PML was restored. Globally, both cell lines studied suggest a role for PML in HHV-6B integration. However, since integration still occurred in PML KO cells, this indicates that the contribution of PML although facilitating, is not absolutely required for this process. We would surmise that when PML-NBs are present at telomeres, SUMOylation of TRF2 is enhanced, predisposing telomeres to recombination events between the viral telomeric repeats and cellular telomeres. To dismiss the fact that less integration in PML KO cells might result from the skewing of infection toward a lytic cycle, we measured the HHV-6B DNA copy numbers in PML KO relative to WT cells. Results showed that both WT and PML KO U2OS and HeLa LT cells have similar viral DNA copy numbers suggesting that

PML does not modify the course of HHV-6B infection in semi-permissive U2OS and HeLa LT cells (Fig 5B and 5D).

Considering the importance of PML on HHV-6B integration, we hypothesized that localization of IE1 at telomeres might be PML dependent. In PML$^{-/-}$ cells, a significant but not total reduction of IE1 at telomeres was observed (Fig 6D). The residual association of IE1 with telomeres can be explained by SUMOylated IE1 interacting with SIM containing proteins present at telomeres. Alternatively, IE1 might reach telomeres through interactions between its putative SIM site and SUMOylated proteins implicated in DNA repair that are found at telomeres, such as the BLM, RAD52 or RPA proteins [72–77].

Our study is not without limitations. As it was the case in a study from another group, we were not able to make or maintain stable clones expressing PML-I [78]. Since viral integration occurs promptly following infection in semi-permissive cells, we had to resort to transient PML-I expressing cells. However, because the expression of PML-I was lost in time, we could not do IF-FISH analysis to compare if the integrated genome was in fact in the U2OS PML + restored cells. One way to improve, although quite challenging, would be to use recombinant viruses mutant for SUMOylation and use them in the same cell lines to assess the importance of SUMOylation of IE1 by PML for HHV-6B integration. Lastly, although our results suggest that $^{775}$VVV$^{777}$ represents a functional SIM site, experiments designed to functionally prove this were not performed.

In conclusion, we have demonstrated that HHV-6B IE1 protein is SUMO-modified at PML-NBs and colocalizes with PML and host telomeres. Abrogation of PML expression abolished multiSUMOylation of IE1, reduced its localization at telomeres and severely impaired HHV-6B integration into host chromosomes. To our knowledge, this is the first report describing a role of a non-shelterin cellular protein involved in HHV-6B integration.

## Material and methods

### Cell lines and virus

MOLT-3 (ATCC, Manassas, VA, USA) were cultured in Roswell Park Memorial Institute (RPMI-1640; Corning Cellgro, Manassas, VA, USA) supplemented with 10% Fetal Bovine Serum (FBS) (Thermo Fisher Scientific, Waltham, MA, USA), HEPES, sodium pyruvate (Wisent Inc., St-Bruno, Québec, Canada), and 5 μg/ml plasmocin (Invivogen, San Diego, CA, USA). Hela LT (39) and HEK293T (ATCC) were cultured in Dulbecco's Modified Eagle's Medium (DMEM; Corning Cellgro, Manassas, VA, USA) supplemented with 10% Fetal Bovine Serum (FBS) (Thermo Fisher Scientific), nonessential amino acids (NEM) (Corning Cellgro), HEPES, sodium pyruvate (Wisent Inc.), and 5 μg/ml plasmocin (Invivogen). U2OS (osteosarcoma) cells (ATCC) and U2OS-Flp-In TREX (kind gift from Dr. Jakob Nilsson, University of Copenhagen) were cultured in the same medium but supplemented with 10% of Nu Serum (Corning Cellgro) instead of FBS and U2OS-Flp-In TREX were maintained with 5 μg/ml of blasticidin (Invivogen). HHV-6B strain Z29 [79] was produced by our laboratory, as previously described [21].

### Plasmids

Expression vectors for HHV-6B IE1 (pcDNA4/TO-IE1B) and control vector (pcDNA4/TO) were described previously (40). Expression vector for HHV-6A IE2 (pcDNA4-IE2) was described previously [80]. Plasmids expressing pCS3-MT-MYC-PML isoforms were kindly provided by Jin-Hyun Ahn [81]. Expression vectors for SUMO paralogues (pCMV2N3T-HA-SUMO-1/ SUMO-2 and SUMO-3) were previously described [22]. To generate a PML-I lentiviral vector, the PML-I gene was PCR amplified with *attB1* and *attB2* sites added to the forward and reverse

primers, respectively. The PCR amplicon was recombined into pDONR221 vector followed by a second recombination into pDEST-CMV Hygro vector (RRID:Addgene_17454), a kind gift from Eric Campeau and Paul Kaufman [82]. Mutation of the RING domain of PML-I in pLenti-CMV-Hygro-PML-I was generated with Q5 site-directed mutagenesis kit by changing the leucine at position 73 for a glutamate (pLenti-CMV-Hygro-PML-I:L73E). Mutation of SUMO site at K802 of IE1 was previously described [21]. SUMO Interacting Motif (SIM) sites of IE1 [573]VVV[575] and [775]VIV[777] were generated in pcDNA4/TO-IE1B and in pcDNA4/TO-IE1: K802R, using Q5 site-directed mutagenesis kit (New England Biolabs, Whitby, Ontario, Canada) by changing the amino acids VVV or VIV for three alanines ([573]AAA[575] and [775]AAA[777]). The PML Double Nickase Plasmids (h2) (sc-400145-NIC-2) were bought from Santa Cruz Biotechnology (Santa Cruz, CA, USA) (S5 Fig). Expression vector pcDNA4-MYC-IE1B was previously described [83].

## MOLT-3 infection assay

$5 \times 10^6$ cells were infected or not (NI), at a MOI of 0.1 with HHV-6B-Z29 for 5 hours in a 15 ml tube at 37°C, 5% $CO_2$. Cells were pellet and washed three times with PBS 1X. The pellets were resuspended in 5 ml of fresh complete RPMI media and incubated in a 25 cm$^2$ flask for 24h and 72h. Upon collection day, 1 ml was taken from each flask, cells were pellet and washed 3 times in phosphate-buffered saline (PBS) 1X. Cells were then counted and put at a final concentration of $10 \times 10^6$ cells/ml. 10 µl was added to a microscope slide with reaction wells. Once dried, cells were fixed for 10 minutes at -20°C in 100% acetone. Dried slides were then kept at -20°C until immunofluorescence staining.

## Transfection assays

U2OS cells were seeded at $2 \times 10^5$ cells/well in a 6-well plate containing five glass coverslips in 2 ml of medium. Cells were transfected 24 hours post-seeding with 2.5 µg of pcDNA4/TO or pcDNA4/TO-IE1B expression vector using the *TransIT*-LT1 Transfection Reagent (Mirus Bio LLC, Madison, WI, USA). After 48 hours of transfection, cells were washed 3 times with PBS, fixed in 2% paraformaldehyde/PBS for 10 minutes at room temperature, washed 10 minutes with PBS and processed for immunofluorescence assay. HeLa LT cells were seeded at $1 \times 10^5$ cells/well in a 6-well plate containing five glass coverslips in 2 ml of medium. Cells were transfected 24 hours post-seeding with 4 µg of pcDNA4/TO, or pcDNA4/TO-IE1B expression vector using Lipofectamine 2000 (Thermo Fischer Scientific). After 48 hours of transfection, cells were fixed as described above and used for immunofluorescence.

## Immunofluorescence (IF)

Fixed cells on coverslips were incubated for 30 minutes in blocking solution (1 mg/ml BSA; 3% goat serum; 0.1% Triton X-100; 1 mM EDTA pH 8.0, in PBS). After blocking, coverslips were incubated for 1 hour in primary antibody diluted in blocking solution. Coverslips were washed with PBS, three times for five minutes and were incubated for 30 minutes with secondary antibody diluted in blocking solution. Next, coverslips were washed three times for 5 minutes in PBS. Coverslips were air dried at room temperature, kept in dark and mounted with *SlowFade* Gold Antifade reagent containing DAPI (Invitrogen, Eugene, Oregon USA).

## Foci volume quantification

Image stacks were open in Fiji from ImageJ (Fiji, RRID:SCR_002285). Each nucleus was selected with a region of interest (ROI). Next, stack for each channel of the ROI were

duplicated and background in the nucleus was subtracted. For each nucleus, the ROI area was measured. Foci volume for each nucleus was quantified with the 3D object counter plugin. To scale up the foci volume to the nucleus size, the area of a nucleus with an empty was used to normalize each foci of every nucleus according to their respective nucleus area.

### Foci number counts

Data generated from foci volume quantification were used to quantify the number of foci per nucleus. For each nucleus, different foci volume data were generated, representing the number of foci within that nucleus.

### Colocalization quantification

Images were first analyzed manually with PerkinElmer Volocity 5.4 software. Each antibody was tested with every laser channel to be sure that the signal was antibody specific (S2 Fig). Colocalization was manually quantified by counting the number of IE1B foci colocalizing with PML, in a 3D (X, Y, Z) manner by going through the z stack of the images for all foci (S2B Fig). Manually counted colocalization was then confirmed with ImageJ software JACoP. Image stacks were open in Fiji. Each nucleus was selected with a ROI and background was deleted throughout the stack of every channels. Next, stack for each channel of the ROI were duplicated and background in the nucleus was corrected to avoid false colocalization. A Costes auto threshold was applied to avoid manual subjectivity. Colocalization was quantified into Mander's colocalization coefficient (MCC). Data of MCC can be interpreted as the proportion of the dye A that colocalize with dye B and vice versa. Indicated values are between 0 and 1 where 1 stands for perfect colocalization.

### BioID stable cell lines

pDEST-cDNA5-BirA-FLAG-N-term and pcDNA5-BirA-FLAG-GFP were kindly provided by Dr. Anne-Claude Gingras (University of Toronto) [84,85]. To be able to generate a stable cell line expressing BirA-IE1 of HHV-6B, the IE1 gene was PCR amplified with *attB1* and *attB2* sites added to the forward and reverse primers, respectively. The PCR amplicon was recombined into pDonor221 vector followed by a second recombination into the pDEST-pcDNA5-BirA-FLAG-N-term now called pcDNA-BirA-FLAG-IE1B.

Stable cell lines were generated by seeding U2OS-Flp-In-TREX cells ($2 \times 10^5$ cells/ well of a 6-well plate). The next day, 277 ng of pcDNA5-BirA-FLAG-GFP or pcDNA-BirA-FLAG-IE1B together with 2.22 µg of the Flp-recombinase pOG44 from Invitrogen (V6005-20) (ratio 1:9, respectively) were co-transfected using the *TransIT*-LT1 Transfection Reagent. 24 hours post transfection, cells were washed, and fresh medium was added. 48h post-transfection, cells were splited at 25% confluence, 250 µg/ml of hygromycin and 5 µg/ml of blasticidin were added. Once selected, cells were then expanded for BioID assay as described by Roux and *al* (80). Briefly, cells were seeded in 10–15 cm dishes as well as seeded on coverslips for IF. Upon 80% confluency, 1 µg/ml of doxycycline and 50nM of biotine were added for 24 hours. For IF, cells were fixed with paraformaldehyde 2% and labelled for BirA-GFP and BirA-IE1B expression (Flag) and biotinylated proteins (Streptavidin-HRP-594). For Mass Spectrometry, cells were harvested, lysed and sonicated with buffer as described (80). Biotinylated proteins were immunoprecipitated with streptavidin magnetic beads and resuspended in 50 mM ammonium bicarbonate. Samples were then analyzed by the mass spectrometry platform. Using the scaffold 4 program, IE1 specific interactors were analyzed by doing a quantitative analysis from a Fisher's Exact Test.

## Co-immunoprecipitation (Co-IP)

HEK293T cells were seeded at a density of 500 000 cells/well in a 6-well plate with 2 ml of medium. 24 hours post-seeding, cells were co-transfected with 2 or 3 μg of total DNA depending on the condition, with a ratio of 3:1 of polyethylenimine at 1 μg/μl (PEI):DNA, in a total volume of 200 μl of DMEM without any serum and antibiotics. Transfection reactions were incubated for 15 minutes at room temperature and added drop-wise in the respective wells. 48 hours post-transfection, cells were harvested. Pellets were resuspended in 500 μl of RIPA lysis buffer (50 mM Tris-base, 150 mM NaCl, 1 mM EDTA, 0.1% Sodium Dodecyl Sulfate (SDS), 0.5% sodium deoxycholate, 1% NP-40) containing 1X Halt Protease Inhibitor Cocktail (100X) (Thermo Fisher Scientific, Waltham, MA, USA) and 10 mM of N-Ethylmaleimide (NEM), and were lysed at 4˚C for 30 minutes under rotation. Lysates were centrifuged for 10 minutes at 12 000 x g, at 4˚C. Supernatants were collected in a new tube. 10% (50μl) of each sample was kept for the input analysis by lysis in 10 μl of 5X loading buffer (0.5 M Tris-HCl pH6.8, 500 mM dithiothreitol, 350 mM SDS, 7.5 mM bromophenol blue, 50% glycerin) and conserved at -20˚C until western blot analysis. The rest of the lysate was used for immunoprecipitation. 2 μg of polyclonal rabbit antibody IE1 was added in each sample and they were incubated for 2 hours at 4˚C under rotation. After 2 hours, protein A/G agarose beads (1:1) were added to each sample an incubated overnight at 4˚C under rotation. Samples were washed three times in RIPA lysis buffer, resuspended in 100 μl of 2X loading buffer (0.5 M Tris-HCl pH6.8, 15% ß-mercaptoethanol, 1.2% SDS, 0.5% bromophenol blue, 30% glycerol) and boiled for 5 minutes at 95˚C, as well as input samples, for western blot analysis (S3 Fig).

## Generation of PML Knockout (KO) cell line

U2OS and HeLa LT cells were transfected with CRISPR-Cas9 and guide RNA expressing vector targeting PML (S5 Fig). After 48 hours, cells were selected with 1 μg/ml of puromycin. Selected cells were harvested, counted and seeded at a density of 1 cell per well in three 96-well flat-bottom plates. After 10 to 14 days, wells containing only a single clone were identified (S5 Fig). Clones were propagated for an additional 3 weeks and transferred into wells of a 12-well plate. Clones were screened by PCR, sequenced and analyzed by IF for PML expression. PML negative clones were expanded and kept frozen until used.

## HHV-6B integration assays

Integration assays were performed as described previously (43) (S6 Fig). Briefly, ten thousand cells per well (U2OS PML WT, U2OS PML$^{-/-}$ #1, U2OS PML$^{-/-}$ #2, HeLa LT PML WT, HeLa LT PML$^{-/-}$ #1, HeLa LT PML$^{-/-}$ #2 and PML restored) were seeded in 48-well plates. The next day, cells were infected with HHV-6B-Z29 at a multiplicity of infection (MOI) of 1 followed by overnight incubation at 37˚C. Cells were washed three times with 1X PBS to remove unabsorbed virions prior to the addition of fresh culture medium. Upon infection, cells were passaged for 4 weeks and DNA was isolated using the QIAamp DNA Blood Mini Kit as described by the manufacturer (Qiagen Inc., Toronto, ON, Canada). The integration frequencies were determined by ddPCR as described (Gravel et al 2017). The HHV-6B chromosomal integration frequencies were estimated assuming a single integrated HHV-6/cell and calculated with the following formula: (number of HHV-6 copies)/(number of RPP30 copies/2 copies per cell) × 100, as previously described [33]. This assay was previously extensively validated and provide comparable data to single-cell cloning and quantification [33]. A chi-square statistical test was done to compare total copy numbers between each condition as described below.

## qPCR

qPCR was performed as described previously by Gravel et al. (43). Briefly, DNA was extracted using QIAamp DNA Blood Mini Kit as described by the manufacturer (Qiagen Inc.) and analyzed using primers and probes against *U65-66* (HHV-6B) and *RPP30* (reference gene). Data were normalized against the corresponding genome copies of the cellular *RPP30* gene.

## U2OS infection assay

U2OS cells were seeded at $2 \times 10^5$ cells/well in a 6-well plate containing five glass coverslips in 2 ml of medium. Cells were infected 24 hours post-seeding at a multiplicity of infection (MOI) of 0.5 with HHV-6B-Z29. At 48 hours post-infection, cells were washed 3 times with PBS and fixed in 2% paraformaldehyde/PBS as described above and used for immunofluorescence assay.

## Immunofluorescence conjugated to *in situ* hybridization (IF-FISH)

Fixed cells on coverslips were stained as for IF. Once IF was completed, cells were fixed for 2 minutes at room temperature with 1% paraformaldehyde/PBS. Coverslips were washed two times for five minutes with PBS. Cells were dehydrated for 5 minutes in successive ethanol baths (70%, 95%, 100%). Once dried, coverslips were placed upside down on a drop of hybridizing solution (70% formamide; 0.5% blocking reagent; 10 mM Tris-HCl pH 7.2; 1/1000 Cy3 or Cy5-TelC PNA probe). Sample were denatured for 10 minutes at 80°C on a heated block. Coverslips were incubated over night at 4°C and kept in the dark. After hybridization, coverslips were washed two times for 15 minutes in washing solution (70% formamide; 10 mM Tris-HCl pH 7.2) and then washed 3 times for 5 minutes with PBS. Sampled were air dried, mounting media was added and coverslips were sealed.

## Statistical analysis

Unpaired t-test was used to compare the foci levels, foci counts, MCC and viral DNA copies. For integration analysis, to total number of cells with integrated HHV-6B were compared to the total number of cells, analyzed using a Fisher's exact test. For the comparison of the patterns of IE1, a Chi-square test was done by comparing the total IE1+ nuclei with non-IE1 + nuclei.

## Supporting information

**S1 Fig. IE1 colocalizes with all PML nuclear isoforms.** (A) Illustration of the main features of the PML protein and its six nuclear isoforms after alternative splicing. The green and blue colors represent exons and the grey color represents introns. (B) U2OS PML$^{-/-}$ were co-transfected using pcDNA4TO-IE1B along with vectors expressing the various Myc-tagged PML isoforms (I to VI). 48 hours post-transfection, cells were analyzed by IF using anti-Myc ALEXA-488-labeled (green) and anti-IE1 ALEXA-568-labeled (red) antibodies. (C) HEK293T were transfected with Myc-tagged PML isoforms expression vectors and analyzed by western blot using anti-Myc antibody.
(TIF)

**S2 Fig. Antibody test and IF-FISH example of colocalization of IE1 with PML at telomeres.** (A) Controls for antibody specificity and fluorescence cross-leakage. U2OS pcDNA4/TO-IE1B transfected cells were labeled separately for anti-PML (green), anti-IE1 (red) and telomeric probe (aqua). Acquisitions were made for each sample to determine whether leakage of

fluorophores occurred. All three filters were specific and did not allow fluorescence leakage. (B) Images representing fluorescence of each target protein/DNA in a X, Y, Z manner (3D) (C) Images representing colocalization of IE1 with PML, at telomeres in 3D (orange square). Combination for each channel can be observed. Triple colocalization gives white foci. (TIF)

**S3 Fig. SUMOylation assay and search for potential SIM sites.** (A) SUMOylation assay protocol in brief. Total cellular lysate is first incubated with IE1 antibody for 1 hour after which Protein A/G agarose beads are added and incubated overnight. After several washes, samples are boiled at 100˚C in 2X laemmli sample buffer for 5 minutes. Samples are separated by a 6% SDS-PAGE and SUMOylated IE1 detected using an anti-HA antibody (for HA-SUMO-1 detection on IE1). (B) The amino acid sequence of IE1 was screened (Q77PU6 from UniProt) with the SUMO and SIM site predictor GPS-SUMO (http://sumosp.biocuckoo.org/online. php). Potential SIM sites are indicated by the rectangles based on the calculated p-values. (TIF)

**S4 Fig. PML localizes at telomeres in ALT+ and HeLa LT cells.** (A) U2OS cells (ALT[+]) and HeLa LT cells (telomerase[+]) were grown on coverslips and fixed with 2% paraformaldehyde at sub confluence. Cells were analyzed by IF-FISH. The PML protein was detected using an anti-PML with an anti-mouse-ALEXA-488 (green) antibodies and telomeres were detected using a Cy3-labeled telomeric probe (red). (B) Graph representing mean±sd of the percentage of PML foci localizing at telomeres in U2OS (N = 20) and HeLa LT (N = 40) nuclei. 71.7%±3.01 of PML colocalize at telomeres in U2OS cells and 40.18%±2.92 in HeLa LT cells. (TIF)

**S5 Fig. PML KO cell line procedures.** (A) Double nickase plasmid backbone. One plasmid is for a first guideRNA in the leading strand of exon 1 of PML, the expression of a Cas9n(D10A) and the puromycin resistance gene. The second plasmid has a guideRNA in the lagging strand overlapping with the first guideRNA, the expression of a Cas9n (D10A) and GFP. (B) Scheme of the double nickase cuts. (C) Schematic representation of the *PMLI* gene indicating where in exon 1 the deletion is introduced. Black arrows represent the primers used for PCR amplification. (D) *PML* gene translation before deletion. Sequence in blue represents the length of the protein after deletion of a part of a sequence in exon 1. Truncated protein is from 1–222 amino acids. Sequence in yellow represents the antigen recognition sequence (aa 31–57) by the monoclonal mouse anti-PML antibody used and purchased from Santa Cruz Biotechnology Inc (SC-966). (E) Experimental procedure used for the generation of PML KO cells. Plasmids expressing Cas-9 and the guide RNAs targeting exon 1 of the *PML* gene were transfected in U2OS and HeLa LT cells. 48 hours post-transfection, cells went under puromycin selection for a week. Selected cells were then plated into single cell per well in 96-wells plate to do a single cell cloning assay. Wells were screened every week for the presence of a unique colony. Once single cell clones were amplified, they were next transferred into more voluminous wells to amplify the clones and screened. (F) PCR amplifications of WT U2OS and HeLa LT cells and their clones with primers designed as in S5C. When mutated, the PML amplification band is at 136bp instead of 198bp, as observed for U2OS clones. WT and mutant bands were extracted and sequenced. For HeLa LT cells, because this cell line contains more than one chromosome 15 (chromosomal location of PML), these clones had more than one amplification band and were therefore screened by PCR and IF. We have selected clone 1 and 2 for each of the cell line for further experiments. (G) Chromatogram showing the deletion introduced by CRISPR-*Cas9* and the *PML* gene translation after deletion within exon 1. (TIF)

**S6 Fig. Integration assay.** Scheme representing the integration assay as previously described by Gravel et al. (36). Briefly, cells were infected at a MOI of 1 for 24 hours and kept in culture for 4 weeks. At day 5, cells were transferred to a bigger culture vessel and a portion of each condition were screen by qPCR to assure HHV-6B presence. HHV-6B copies were detected with *U65-66* primers and normalized with the cellular reference gene (*RPP30*). (B) Scheme showing the results of an inherited chromosomally integrated HHV-6B sample estimated integration rate by ddPCR. Briefly, cell's DNA together with a Taq polymerase and primers with their specific fluorescent probe, for instance, are divided into 20 000 droplets of oil. These droplets are then PCR amplified for both of the target genes (*U65-66* and *RPP30*). After the amplification, each droplet is quantified for their number of copies for each gene. (C) Example of calculations to determine the integration rate of HHV-6B. For statistical purposes, the total number of cells with integrated HHV-6B and cells without integration were used. The number of cells is estimated using the *RPP30* gene, present at 2 copies/cell. A Fisher's exact test is done with these values.
(TIF)

**S7 Fig. IE1 localizes at telomeres in HeLa LT cells.** (A) IF-FISH of transfected HeLa LT with pcDNA4/TO-IE1B. 48h post-transfection cells were fixed and labelled for IE1 (red), PML (green) and telomeres (aqua). (B) Graph representing the level of colocalization of IE1 with PML. (C) Graph representing the level of IE1 that localizes at telomeres.
(TIF)

## Acknowledgments

We thank Julie-Christine Lévesque for technical help with the confocal microscopy analyses and thank the Bioimaging platform of the Infectious Disease Research Centre.

## Author Contributions

**Conceptualization:** Vanessa Collin, Annie Gravel, Louis Flamand.

**Data curation:** Vanessa Collin.

**Formal analysis:** Vanessa Collin, Louis Flamand.

**Funding acquisition:** Louis Flamand.

**Methodology:** Vanessa Collin, Annie Gravel, Louis Flamand.

**Resources:** Benedikt B. Kaufer.

**Supervision:** Louis Flamand.

**Writing – original draft:** Vanessa Collin, Louis Flamand.

**Writing – review & editing:** Vanessa Collin, Annie Gravel, Benedikt B. Kaufer, Louis Flamand.

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
