## [Decision Letter · Decision Letter 0]

20 May 2020

Dear Dr. Flamand,

Thank you very much for submitting your manuscript "PML is a SUMO ligase for Human Herpesvirus 6B Immediate Early 1 protein that facilitates viral integration" for consideration at PLOS Pathogens. As with all papers reviewed by the journal, your manuscript was reviewed by members of the editorial board and by several independent reviewers. The reviewers appreciated the attention to an important topic. Based on the reviews, we are likely to accept this manuscript for publication, providing that you modify the manuscript according to the review recommendations.

Dr Flamand,

I hope you are safe and sound. Your manuscript has been evaluated by three expert herpesvirologists who are all highly supportive of your work. To your credit, they note considerable changes to the manuscript from the first offering that reflect a significant improvement and serious response to the original reviews. They do note some interpretation issues and request additional clarity concerning the conclusions and analysis of the experiments which I believe can be easily accommodated by a modification. If you choose to modify the manuscript, I would be more than happy to evaluate the manuscript myself. If there are any questions, please feel free to reach out to me. Cheers,

Eain Murphy

Sincerely,

Eain A Murphy, Ph.D.

Associate Editor

PLOS Pathogens

Shou-Jiang Gao

Section Editor

PLOS Pathogens

Kasturi Haldar

Editor-in-Chief

PLOS Pathogens

orcid.org/0000-0001-5065-158X

Michael Malim

Editor-in-Chief

PLOS Pathogens

orcid.org/0000-0002-7699-2064

Dr Flamand,

I hope you are safe and sound. Your manuscript has been evaluated by three expert herpesvirologists who are all highly supportive of your work. To your credit, they note considerable changes to the manuscript from the first offering that reflect a significant improvement and serious response to the original reviews. They do note some interpretation issues and request additional clarity concerning the conclusions and analysis of the experiments which I believe can be easily accommodated by a modification. If you choose to modify the manuscript, I would be more than happy to evaluate the manuscript myself. If there are any questions, please feel free to reach out to me. Cheers,

Eain Murphy

Reviewer Comments (if any, and for reference):

Reviewer's Responses to Questions

**Part I - Summary**

Reviewer #1: This study addresses the interaction between the HHV-6B IE1 and the human PML protein, effects of PML on IE1 SUMOylation, and effects of SUMOylation on IE1 subnuclear distribution. The localization of IE1 and PML at telomeres and the role of PML in HHV-6B integration into the host chromosomes is also investigated. The finding that PML may promote HHV-6B chromosomal integration is of considerable importance. The data and analysis are substantially improved compared to the previous manuscript version I reviewed. Especially the microscopy-based quantitative analyses are overall very convincing. However, the study is flawed by a number of overinterpretations. The conclusions are generally consistent with the data, but there are alternative interpretations for many of the results that have not been properly considered.

Reviewer #2: The mechanisms that facilitate the integration of HHV-6A and HHV-6B into the telomeres of the host chromosome are unknown in the field. Collins et al. report several novel observations that indicate PML associates with the immediate early protein (IE1) of HHV-6B at nuclear foci, functions as a SUMO ligase for IE1, and facilitates the association of IE1 with telomeres. PML also facilitates the integration of HHV-6B. While the authors did not make the causal link of PML SUMOylation to integration, they have generated a plentiful amount of robust data that supports the generation of an intriguing hypothesis. This submission has addressed all initial concerns of this reviewer with the addition of immunoprecipitation data, quantitation of immunofluorescence, details of the methods and analyses, and technical controls. The most significant addition is the data from Figures 4 and 5 that demonstrate PML is a SUMO ligase for IE1, leading to hyperSUMOylation via SUMO-1; they identified a SIM site of IE1 that influences IE1/PML foci. Grammatical errors remain; editing is needed to improve readability.

Reviewer #3: This is an interesting study, which has been improved in its revised version. Understanding the mechanisms of HHV-6B integration is clearly of importance.

Nevertheless, there are still some aspects that require attention.

I my reading, the title does not cover exactly what has been shown in the paper. They show that PML induces sumoylation of IE1, that IE1 associates with PML NB, that PML and IE-1 are colocalized with telomeres and that a lack of PML negatively effects HHV-6B integration. My reading of the title is that the PML effects on IE1 facilitates viral integration. But it is not demonstrated that PML exerts its effect on integration via IE1 (but that may of course be the case). This conclusion would require the ability to “remove” IE1 from the infected cell.

**Part II – Major Issues: Key Experiments Required for Acceptance**

Reviewer #1: 1. The data show that IE1 is hyper-SUMOylated in the presence of PML, but do not show that PML is a SUMO ligase for IE1. Thus, claims like “IE1 is SUMOylated by PML” (lines 23-24) or “PML act[s] as a SUMO ligase for IE1” (lines 28-29) have to be removed from the text or justified experimentally (e.g. by SUMOylation assays with and without PML and/or assays involving an IE1 mutant selectively deficient for PML binding). The positive effect of PML on SUMOylation may be indirect, as recently proposed for hCMV IE1 (PMID 32365141). This is particularly likely, since the interaction between PML and HHV-6B IE1 may be indirect (Figure 1). Perhaps localization of IE1 at PML bodies enhances SUMOylation irrespective of PML acting as a SUMO ligase? In this context, the authors should also acknowledge that their L73E PML mutant may not be selectively deficient for only the SUMO ligase activity but other PML functions as well.

2. The VIV motif (amino acids 775-777) the authors identified in HHV-6B IE1 may serve an important function, but it is not legitimate to refer to this motif as a SIM (SUMO interacting motif) unless it is shown experimentally that it confers non-covalent interaction with SUMO proteins. Thus, all firm claims that amino acids 775-777 serve as a SIM have to be removed from the manuscript and figures unless the relevant interaction data are provided. In line 215, “773-775” needs to be changed to “775-777”. Instead, it should be clear that the authors are merely speculating that amino acids 775-777 may be part of a SIM. Alternative (or additional) to interacting with SUMO, the VIV motif may be involved in interactions with other proteins. It may, for example, mediate SUMO-independent binding to PML. In hCMV, a similar motif has been implied in interactions with STAT2 and STAT3 (PMID 27387064), and the authors have previously shown that HHV-6B IE1 binds to STAT2 as well (PMID 20404187).

3. The data do not show that “IE1 multi-SUMOylation […] is important for IE1 to localize at telomeres” (lines 342-343). A SUMOylation-deficient IE1 mutant needs to be studied to justify this conclusion.

Reviewer #2: (No Response)

Reviewer #3: 1. An important piece of information is the consequence of PML knock out. The PML knock out cells were apparently tested by PCR (comments to reviewers), please show these data in the supplement figure 5. The cells were, however, tested by IF, but information on the source of anti-PML antibody, including whether it is known to recognize all the isoforms is missing – please provide this information.

Moreover, I noticed that in a comment to one of the reviewers it is stated that TIDE was not performed because it was done by Santa Cruz. This appears to be irrelevant, unless Santa Cruz made the cells. A TIDE analysis is performed on the cells after the CRISPR/Cas attempt to knock out the gene, how can Santa Cruz have performed this?

2. Fig. 5B is difficult to understand. What is PML-/- #1 and #2? What does error bar indicate? (used at several places in figure legend without further explanation – please correct), and the y-axis does not give a hint on what has actually been measured (integration). But more troublesome, it is not clear why a Chi-square test has been used and even more difficult to understand that if the error bars indicate the standard deviation, how can a result on three measurements be significant with a p-value of 0.0008 when the error bar for PML-/- #2 encompasses the normalized value for PML+/+. This needs a careful explanation potentially providing raw data.

**Part III – Minor Issues: Editorial and Data Presentation Modifications**

Reviewer #1: 4. It would greatly strengthen the manuscript if the authors could get some indication of whether HHV-6B IE1 is important for chromosomal integration.

5. The authors should be more careful in interpreting Figure 5. The average differences between PML +/+ and PML -/- cells in panel B are very small, and the error bars are huge. The results in panel E look more convincing, but why is there such a great difference between the two PML -/- cell lines?

6. Given the results in Figure 4, it appears inconsistent that “there are no significant changes in the number of IE1 foci between PML +/+ and PML -/- cells” (lines 333-334 and Figure 6E). Please explain.

7. It should be clearer that the mechanisms illustrated in Figure 3B and Figure 4E are hypothetical and that there are alternative interpretations for the presented data. Figure 7 appears to be dispensable. At least, there appears to be major overlap between Figure 3E and Figure 7A, and Figure 7B is somewhat generic.

8. Does “IE2A” refer to HHV-6A IE2? Why was an HHV-6A protein used as a control alongside HHV-6B IE1?

9. Provide a reference number for “Osterwalds et al” (line 374).

10. The authors have to proof-read their manuscript for spelling mistakes, grammatical errors and other small inconsistencies. There are many of them, even in the Summary (e.g. “Taken together, we could demonstrate that PML act as a SUMO ligase […]”) and Author Summary sections (e.g. “HHV-6B is among few other herpesviruses that integrate its genome in host chromosomes as a mean to establish dormancy”). Other examples include, but are not limited to:

- “what effect it could have on IE1” (line 172);

- “the presence of PML-I:L73E ” (line 202);

- “resulting in larger NBs (Figure 4E)” (line 264);

- “acceptor sites on IE1 (Figure 3D)” (line 241);

- “phenotypes of IE1 mutants” (line 245);

- “two distinct cell types” (line 297);

- “PML deficient cells integrated HHV-6B less frequently” (lines 305-306);

- “were infected at an MOI of 1” (line 308);

- “TRF2 may serve to protect” (line 365);

- “Ubc9 can bind SUMO directly at a SIM” (line 419);

- “since integration still occurred in PML KO cells, PML’s contribution” (lines 444-445);

- What does “sewing” mean in line 447?

Reviewer #2: 1. Subject-Verb tense issues.

a. Line 44: ‘prompt<ed> us’

b. Line 45: ‘that are essentials’

2. Lack of articles.

a. Line 63: ‘Thus far, <the> state’

b. Line 83: ‘Furthermore, <the> integrated’

c. Line 379: ‘role of PML in <the> DNA repair mechanism

3. Unclear/poor wording.

a. Lines 82-81: ‘In support, the integrated…’

b. Lines 205-207

c. Line 223: ‘affected’ is vague, perhaps better described as ‘reduced IE1-SUMO-1 conjugation’

d. Lines 262-264: ‘causes’ and ‘causing’

e. Lines 341-343: This sentence is misleading- wording tries to walk a fine line that blurs the distinction- be clear that no causal link can be made without the IE1 SUMO mutant.

f. Lines 408-411: ‘considering’ and ‘considering’

g. Lines 458-461: run-on sentence

h. Line 467: ‘auto oligomerization’?

4. Use of the possessive for nouns is excessive and not typical for science communications.

Examples are line 113, 170, 245,408,1070,1079, 1081, 1104, 1179, 1180 but found throughout manuscript

5. Lines 122-124- put citation directly after author’s name ‘Stanton et al.’

6. Line 145 correct to ‘we studied the’

7. Typos

a. Line 156 ‘pull downed’

b. Line 157 ‘my mass spectrometry’

c. Line 167 ‘Taking together’

d. Line 184 ‘mono, poly or. ‘

e. Be consistent with using ‘Figure’ or ‘figure’

f. Line 290 ‘3 to 40 folds’

g. Line 376 ‘We’ve’- conjugations should be avoided

h. Line 475 ‘lost <with> time’

i. Line 1079 ‘SUMOylation is enhance’

j. Line 1097, 1140, 1145 and elsewhere ‘means’

k. Line 1114 ‘and processes’

l. Line 1139 and elsewhere ‘folds’ *also should be removed from axis labels in figure 5

m. Line 1143 ‘A qPCR was done’

8. SUMO 1, 2, 3 not detailed in Figure 2B.

9. Bar graph for Figure 2F is a bit oversized and overkill since there is only a low amount of signal for the one condition.

10. Break in paragraph at line 215 not needed.

11. Lines 267-268, specify in U20S cells

12. The rationale for figure 6 is confusing. Clearly state U20S cell are ALT+ at beginning of paragraph.

13. The model in Figure 7 is not a valuable addition to the manuscript.

Reviewer #3: I cannot read from the methods how random (unbiased) selection of nuclei was performed for the confocal data? Given the great variability (also seen in fig. 1B IE foci) how can conclusions of number of foci in time be achieved when only to measurements is performed? (again fig. 1B IE foci at 24 hrs).

Therefore, the statement l.129 “the number of foci diminishes with time” appears not to be supported by the data for IE foci and should be removed, or more data points should be included.

l.30: …by these viruses to maintain their… -> …by this virus to main its…?

l.66: HHV6-B -> HHV-6B

l.86: characterizes -> characterized

l.88: ref 18 is missing volume and page numbers.

l.142: “show perfect colocalization”, and in l. 144 about the same finding: “almost perfect correlation”? Both cannot be true. Please characterize the colocalization in a more scientific manner.

l.184: remove the period after “or.”

  </with></the></the></the></ed>

PLOS authors have the option to publish the peer review history of their article (what does this mean?). If published, this will include your full peer review and any attached files.

Reviewer #1: No

Reviewer #2: No

Reviewer #3: No
---

## [Editor Report · Decision Letter 1]

4 Jun 2020

Dear Dr. Flamand,

We are pleased to inform you that your manuscript 'The Promyelocytic Leukemia Protein facilitates human herpesvirus 6B chromosomal integration, immediate-early 1 protein multiSUMOylation and its localization at telomeres.' has been provisionally accepted for publication in PLOS Pathogens.

Best regards,

Eain A Murphy, Ph.D.

Associate Editor

PLOS Pathogens

Shou-Jiang Gao

Section Editor

PLOS Pathogens

Kasturi Haldar

Editor-in-Chief

PLOS Pathogens

orcid.org/0000-0001-5065-158X

Michael Malim

Editor-in-Chief

PLOS Pathogens

orcid.org/0000-0002-7699-2064

Dear Dr. Flamand,

I have reviewed your manuscript and feel the changes you have made in response to the reviewers comments were sufficiently addressed. I am recommending acceptance of this without further review. Congratulations on a nice piece of work.

Cheers,

Eain Murphy
---

## [Editor Report · Acceptance letter]

1 Jul 2020

Dear Dr. Flamand,

We are delighted to inform you that your manuscript, "The Promyelocytic Leukemia Protein facilitates human herpesvirus 6B chromosomal integration, immediate-early 1 protein multiSUMOylation and its localization at telomeres.," has been formally accepted for publication in PLOS Pathogens.

Best regards,

Kasturi Haldar

Editor-in-Chief

PLOS Pathogens

orcid.org/0000-0001-5065-158X

Michael Malim

Editor-in-Chief

PLOS Pathogens

orcid.org/0000-0002-7699-2064